

# Page curves and bath deformations

Elena Caceres[1⋆], Arnab Kundu[2†], Ayan K. Patra[2‡] and Sanjit Shashi[1∘]

**1** Theory Group, Department of Physics, University of Texas, Austin, TX 78712, USA
**2** Theory Division, Saha Institute of Nuclear Physics, HBNI,
1/AF Bidhannagar, Kolkata 700064, India

⋆ elenac@utexas.edu , † arnab.kundu@saha.ac.in ,
‡ ayan.patra@saha.ac.in , ∘ sshashi@utexas.edu

## Abstract

We study the black hole information problem within a semiclassically gravitating $AdS_d$ black hole coupled to and in equilibrium with a $d$-dimensional thermal conformal bath. We deform the bath state by a relevant scalar deformation, triggering a holographic RG flow whose "trans-IR" region deforms from a Schwarzschild geometry to a Kasner universe. The setup manifests two independent scales which control both the extent of coarse-graining and the entanglement dynamics when counting Hawking degrees of freedom in the bath. In tuning either, we find nontrivial changes to the Page time and Page curve. We consequently view the Page curve as a probe of the holographic RG flow, with a higher Page time manifesting as a result of increased coarse-graining of the bath degrees of freedom.

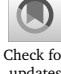
# 1   Introduction

Recent progress has given us a new way to think about the black hole information paradox [1], a central question in quantum gravity. The key insight is that in gravitational systems coupled to an *external bath*, the fine-grained entropy of the Hawking radiation going into a "radiation region" $\mathcal{R}$ is given by the *generalized entropy* $S_{\text{gen}}$, which includes contributions from bulk degrees of freedom in "islands" $\mathcal{I}$ [2–5].

$$S(\mathcal{R}) = \min_{\mathcal{I}} \text{ext}\, S_{\text{gen}}(\mathcal{R} \cup \mathcal{I}), \tag{1}$$

$$S_{\text{gen}}(\mathcal{R} \cup \mathcal{I}) = \frac{A(\partial \mathcal{I})}{4 G_d} + S_{\text{matter}}(\mathcal{R} \cup \mathcal{I}). \tag{2}$$

Under this prescription, we minimize $S_{\text{gen}}$ counting both quantum and gravitational degrees of freedom [6], including possible subregions $\mathcal{I} \subset \mathcal{M}_d$ which are treated as redundant with $\mathcal{R}$. The $\mathcal{I}$ minimizing the entropy is the *entanglement island*. By accounting for the emergence of such an island, one finds a Page curve consistent with unitary evolution of the black hole.

The island rule has been studied so far in a variety of toy models far removed from standard Einstein gravity. The richest chapter of the story has been in 2-dimensional dilatonic gravity, with the island rule being obtained from replica wormholes [7] and thermodynamic tools [8,9] only readily available in 2 dimensions. Higher-dimensional ($d > 2$) toy models have been constructed by embedding (into AdS$_{d+1}$) braneworlds which localize gravity as in the Karch-Randall-Sundrum construction [10–12]. Such models are called "doubly holographic" because they have three equivalent descriptions (see Section 1.1). This construction features both a nongravitating external bath (the conformal boundary) coupled to the brane and entanglement islands [13–16], but gravity on the brane is massive because of the bath [17]. This brings into question the physicality of even having a nongravitating bath in the first place in such higher-dimensional braneworld models, i.e. whether theories with a bath can truly impart lessons about black holes in our own universe.

The underlying theme of this critique is that, in the higher-dimensional models, the bath gives too much computational control over the picture—a satisfactory toy model should not have such a bath in the first place. One way to demonstrate this effect would be to ask how the introduction of a dimensionful scale may affect physical quantities characterizing black hole entanglement dynamics. To this end, we deform the bath theory by a relevant[1] operator, which introduces a new, tunable scale in the bath and thus breaks conformal invariance. The usual logic of double holography—that $(d + 1)$-dimensional *classical* geometry describes the

---

[1]While one may also consider an irrelevant deformation, the corresponding Klein-Gordon potential in our minimal setup will not satisfy swampland bounds [18].

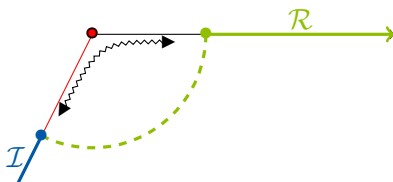

Figure 1: A cartoon depiction of the setup, with the brane in red, the bath in black, and the interface represented as a dot. The two-headed arrow indicates transparent boundary conditions. By computing the entanglement entropy of the radiation region $\mathcal{R}$, the island rule demands that we *also* minimize over possible regions on the brane $\mathcal{I}$. Classically in (III), we end up with the $\mathcal{I}$ for which the entanglement surface area (possibly including brane-action contributions) is minimal.

semiclassical Page curve in the braneworld—hints that this sort of bath deformation will indeed influence the Page curve because it would correspond to bulk classical backreaction.

We use an eternal two-sided black hole—which features an eternal version of the information paradox [19]—as our test bed and study how deformations of the bath corresponding to a scalar field $\phi$ in the bulk affect the Page curve. While the size of $\mathcal{R}$ provides us with one parameter by which to tune the Page curve (specifically its saturation entropy), the deformation triggers a holographic RG flow [20–23] from a UV fixed-point state on the boundary to an IR state at the horizon, then to an analytically-continued *"trans-IR"* flow [24].[2] The "strength" of this deformation (i.e. its boundary source term) provides us with a scale that may be dialed arbitrarily to change the Page time. Thus, the bath is not a consistent computational tool.

## 1.1 Double Holography

We focus on a *doubly holographic* setup [13]; a class of models where the island rule can be written holographically. These systems have three equivalent descriptions:

(I) a $d$-dimensional boundary conformal field theory (BCFT), i.e. a $d$-dimensional CFT with a $(d-1)$-dimensional boundary [25, 26],

(II) a $d$-dimensional CFT coupled to gravity on an asymptotically $\text{AdS}_d$ space $\mathcal{M}_d$, with a half-space CFT bath coupled to $\mathcal{M}_d$ via transparent boundary conditions at an interface point,

(III) Einstein gravity on an asymptotically $\text{AdS}_{d+1}$ space containing $\mathcal{M}_d$ as an "end-of-the-world" brane [10–12].

A particularly useful manifestation of double holography is when the end-of-the-world brane is "tensionless" in the sense of the Karch-Randall-Sundrum constructions [10–12]. While such a "probe" brane does not backreact on the bulk geometry of (III), there is still a tower of spin-2 Kaluza-Klein (KK) modes living on the brane [11]. As discussed in [17], one may still consider the picture (II) by taking the lowest-mass mode to be a graviton and the higher modes to compose the CFT.[3] While such a theory is certainly not standard Einstein gravity, the upshot of using a tensionless braneworld is that holographic calculations in the bulk (III) do not require particularly intricate numerics, unlike in setups with nontrivial tension parameters [13,16,27].

---

[2]In other words, the radial coordinate—identified with the energy scale of the holographic RG flow—becomes timelike. We thank Sean Hartnoll for clarifying this point.

[3]We elaborate on this point in Section 3.

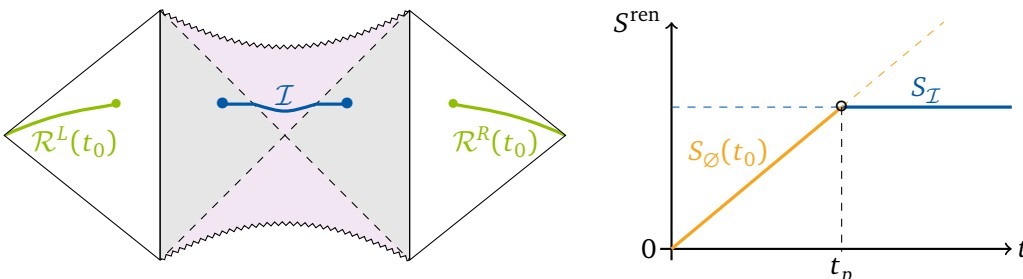

Figure 2: On the left is the two-sided thermal configuration of (II) featuring both the $t = t_0$ radiation region $\mathcal{R}^L(t_0) \cup \mathcal{R}^R(t_0)$ and the island $\mathcal{I}$ which emerges at late times. The right figure is a sketch of the (eternal) Page curve depicting (renormalized) entanglement entropy versus time—initially the time-dependent no-island entropy $S_\varnothing$ is minimal, but it eventually exceeds the non-trivial island entropy $S_\mathcal{I}$. This curve can be obtained by computing (renormalized) areas in the (III) configuration.

The scenario relevant for the black hole information paradox is (II). The relationship between (I) and (III) is the *AdS/BCFT correspondence* [28, 29]. The advantage of such doubly holographic models is that the interesting semiclassical physics in (II) can be extracted from computations performed classically in (III) [4, 14, 30, 31]. Concretely, the generalized entropy of (II) is well-approximated, to leading order in $1/G_N$, in (III) by a classical entanglement surface computed via the *Ryu-Takayanagi (RT)* prescription [32] (or its covariant extension [33])—the surface is extremal and thus must satisfy some boundary condition on the brane.[4] This perspective allows us to interpret the island $\mathcal{I}$ of the entanglement surface of the radiation region $\mathcal{R}$ as "completing" the standard homology condition and serving as a portion of the boundary of the full fixed-time entanglement wedge.

We note that interpreting $\mathcal{R}$ as embodying the degrees of freedom of Hawking radiation from some black hole on the brane is simply one interpretation commonly seen in double holography [13–17, 27] which we also employ. One could reasonably argue that, although the entropy of such intervals is well-defined, it is not the appropriate quantity to compute when dealing with black hole information in these setups. However, studying Hawking radiation would then require an alternative entropy proposal altogether.

## 1.2 The "Eternal" Information Paradox

To see how islands appear in doubly holographic braneworlds, we can consider a simpler analog to the information paradox of evaporating black holes which instead appears when examining eternal black holes coupled and in thermal equilibrium with a bath.

In the landscape of double holography, we consider the evolution of two BCFT systems comprising a thermofield double state characterized by inverse temperature $\beta$. The corresponding solution in (II) is a two-sided $AdS_d$ black hole in thermal equilibrium with two finite-temperature baths, while the solution in (III) is an $AdS_{d+1}$ black hole with a brane present. At a given time $t = t_0$, the radiation region of interest (shown in Figure 2) consists of two disconnected pieces—$\mathcal{R}^L(t_0)$ and $\mathcal{R}^R(t_0)$.

In (III), following the RT prescription without regarding the islands—that is for $\mathcal{I} = \varnothing$— tells us that *Hartman-Maldacena surfaces* encode the entanglement entropy of radiation [34].

---

[4]Technically speaking, it is $S_{\text{matter}}$ which is well-approximated by such an area. However, so long as the only gravitational terms on the brane are "induced" by gravity in the bulk, the $G_d^{-1}$ term vanishes at tree level and thus counts as a quantum correction which we neglect in a semiclassical approximation taking only an effective theory on the brane. This is discussed by [14, 27].

In satisfying the homology constraint, these spacelike surfaces cross the interior of the $AdS_{d+1}$ black hole's Einstein-Rosen bridge. The entropy thus exhibits monotonic growth from $t = 0$, with the late-time $t \gg \beta$ growth being linear. This eternal growth is an information paradox in (II)—the Hawking radiation becomes more entropic than even the black hole [19].

This issue is resolved if nontrivial islands are considered. In particular, [19] discusses how the minimal entropy after the *Page time* $t = t_p$ counts an entanglement island which goes outside of the black hole interior, so at $t = t_p$ there is a phase transition. This is explicitly seen by a classical computation in (III) if we compare the Hartman-Maldacena area—a time-dependent quantity—to the area of a minimized extremal surface residing solely outside of the interior and ending on the brane—a time-independent quantity. The resulting late-time entropy is thus constant, as found by [19]. This means that the Page curve is indeed encoded by the semiclassical picture in which gravity resides on the brane and radiation escapes into a bath.

Note that this phase transition depicted in the eternal Page curve (Figure 2) is analogous to the typical Hartman-Maldacena phase transition seen with no branes [34]. With no branes, we have an RT surface which is entirely homologous to the boundary interval and in the exterior, so this surface would bound the late-time entropy and produce the same sort of entropy curve. In fact, the setup with the tensionless brane is precisely a $\mathbb{Z}_2$ orbifold [17, 35], and so the phase transition of interest is literally the one of Hartman and Maldacena when interpreted in the bulk (as opposed to one which is informed by some nontrivial extremal surface boundary condition at the brane).

Nonetheless, the important interpretation is that of the braneworld theory (II), not the bulk theory (III). The former is where we may view this entropy curve as corresponding to a phase transition of quantum extremal surfaces [6], while the latter is simply a classical phase transition.

As an aside, the comparison can be done entirely on the $t = 0$ slice, for which there is no interior contribution to a Hartman-Maldacena surface's area. Fixing the endpoints of $\mathcal{R}^L$ and $\mathcal{R}^R$ to be the same distance from the brane, one can find whether or not the island surface is already minimal at $t = 0$ [17]—this would imply no phase transition and no Page curve. In doing so, one finds that the radiation region cannot be too close to the brane to get a Page curve (see Section 3.1).

## 1.3 Massive Gravity from the Bath

All of this begs the question—to what extent does the coupling of an external bath influence the story? Previous work has explored the effect of the graviton mass. It is known that in $d \geq 4$, by coupling the $AdS_d$ brane to a bath with a transparent boundary condition at the interface, the localized gravitational theory on the brane becomes a theory of massive gravity with no massless graviton [36, 37]. In [17] the authors tuned the near-zero mass down, finding that the nontrivial island candidate surface grows in response. This limit is essentially performed by tuning the tension of the brane up towards its critical value; in response however, the brane's cosmological constant $\Lambda_b$ and dynamical gravity on the brane turns off. A way out which preserves the massless graviton and dynamical gravity was found by [38], which puts the critical brane at a cutoff surface in the $AdS_{d+1}$ bulk to keep $G_d$ finite. This procedure saves the islands for $d = 2$, although whether it works for $d > 2$ remains to be seen.[5]

A different way to introduce a zero-mass graviton [27] is to make the bath itself *gravitating*, but this gives a *constant* entropy curve if we follow a dynamical[6] island rule—a result in

---

[5]The case of $d > 2$ would *also* need to be reconciled with [39], which argues for incompatibility between islands and massless gravitons in $d > 2$ using the Hamiltonian constraints of gravity.

[6]Here, "dynamical" simply means that the higher-dimensional classical island surface has Neumann boundary conditions on both branes.

agreement with [40]. Note however that there is disagreement in the literature on whether the flat curve of [27, 40] actually depicts the appropriate entropy for a black hole in the first place; [41] argues for a different factorization from [40] as being relevant to the radiating black hole while [42] uses the gravitating bath of [27] with alternate (Dirichlet) boundary conditions for the island surface. [41, 42] both find Page curves in their respective analyses, with the presence of a bath potentially muddying the picture of how one should factorize.

Differences in the literature and pending questions aside, a lot can be learned by either slightly deviating from or further probing the typical "brane and nongravitating half-space BCFT bath" construction, as done by [16, 42–47].

## 1.4 Deforming the Bath

In this work we pursue a different way of altering (although not eliminating) the nongravitating bath: we study a bath deformation corresponding to a scalar field in the bulk and explore how it affects the Page curve. With no branes, it is known that introducing a scalar deformation on the conformal boundary of an AdS black hole will change the near-singularity geometry from AdS-Schwarzschild to a *Kasner universe* [24, 48–50]. The Kasner universe metric is essentially that of the Schwarzschild geometry but with some extra warp factors. Up to pre-factors and in $d + 1$ bulk dimensions ($d \geq 2$), the near-singularity geometry and scalar field behave as (restricting to isotropic gravity + scalar solutions),

$$ds^2 \sim -d\tau^2 + \tau^{2p_t} dt^2 + \tau^{2p_x} d\vec{x}^2, \quad \phi(r) \sim -\sqrt{2} p_\phi \log \tau, \tag{3}$$

where $\tau, t \in \mathbb{R}$, $\vec{x} \in \mathbb{R}^{d-1}$, and $p_t, p_x, p_\phi$ are the *Kasner exponents*. These exponents obey a set of constraints,

$$p_t + (d-1)p_x = 1, \tag{4}$$
$$p_\phi^2 + p_t^2 + (d-1)p_x^2 = 1, \tag{5}$$

so there is only one free exponent.

Notably, these exponents affect the *entanglement velocity*—the speed of the late-time linear growth of the Hartman-Maldacena surface spanning the interior. As this is also the early-time entanglement surface when a brane is present, it is natural to think that the Page curve and, in particular, the Page time will also change.[7]

The deformed geometries studied by [24] are called *Kasner flows* because, from the holographic RG flow program [20], the scalar deformation induces an RG flow from a UV fixed point state on the conformal boundary to a late-time singularity in the black hole's interior [51, 52], with the scaling changing from spacelike to timelike at the horizon. Thus as mentioned above, the flow is to an IR state at the horizon then gets analytically-continued to a trans-IR flow towards the Kasner universe.

The flows are labeled by a dimensionless parameter on the boundary. Each of these flows thus describes a particular coarse-graining of the UV state, controlled by a corresponding radial scale $r_{\rm RG}$ which is probed by the Hartman-Maldacena surface. This scale is roughly defined such that the UV physics dominates between the conformal boundary and $r_{\rm RG}$, while the IR/trans-IR physics becomes more important between the singularity and $r_{\rm RG}$. Concretely, we get a more rapidly coarse-grained state as $r_{\rm RG}$ approaches the boundary.

---

[7]There is a subtlety regarding $d = 2$ when adding a brane: the lowest order terms in the brane's induced gravity action are a nondynamical Einstein-Hilbert term and a nonlocal Polyakov $R \log |R|$ term [14]. As we are assuming an effective dynamical description on the brane, any of our statements about Page curves only work for $d > 2$.

Upon adding a brane, each flow becomes one of a BCFT thermal state [53, 54], and we must account for the island surface. This introduces an independent radial scale $r_T$—the intersection depth of the island surface with the brane—which is directly determined by the metric functions and the size of the radiation region $\mathcal{R}$. This radial scale is tied to how many bath degrees of freedom we trace out when directly computing the entropy of Hawking radiation. It is thus related to the "dynamics" of the entanglement entropy; a larger $r_T$ corresponds to tracing-out more degrees of freedom and thus a higher saturation value of the entropy.

From this perspective, when adding a brane the resulting Page curves found by the island rule probe both the coarse-graining and the entanglement dynamics. Purely from a BCFT perspective, we would heuristically expect two things: (1) that strong IR effects in the bath should increase the Page time by decreasing the $c_{\text{bulk}}$ [53] and (2) that the dynamical part of the Page curve can only be seen after integrating away a minimum number of degrees of freedom. We indeed find both to be the case at least within the numerical range we explore.

## 2 Kasner Flows as AdS/CFT Solutions

We start with a review of the Kasner flows, following [24] but generalizing their equations. We take $(d + 1)$-dimensional Einstein gravity ($d \geq 2$) with negative cosmological constant $\Lambda = -d(d-1)/2$ (setting the AdS radius to 1) and coupled to a scalar field $\phi$ with potential $V(\phi)$. With $16\pi G_{d+1} = 1$, the action is,

$$I = \int d^{d+1}x \sqrt{-g}\left(R + d(d-1) - \frac{1}{2}[\nabla^\alpha \phi \nabla_\alpha \phi + V(\phi)]\right). \tag{6}$$

As in [24], we consider the minimal case of a free massive scalar field—the potential being $V(\phi) = m^2\phi^2$. Note that the flows solving this action with an additional higher-order $\lambda\phi^4$ coupling have been studied by [55].

The bulk equations of motion are the usual Einstein + scalar equations and the Klein-Gordon equations (defining $\Box = \nabla_\alpha\nabla^\alpha$),

$$G_{\mu\nu} - \frac{d(d-1)}{2}g_{\mu\nu} = \frac{1}{4}\left[2\nabla_\mu\phi\nabla_\nu\phi - g_{\mu\nu}\left(\nabla^\alpha\phi\nabla_\alpha\phi + m^2\phi^2\right)\right], \tag{7}$$

$$(\Box - m^2)\phi = 0. \tag{8}$$

For the metric, we take solutions of the form,

$$ds^2 = \frac{1}{r^2}\left[-f(r)e^{-\chi(r)}dt^2 + \frac{dr^2}{f(r)} + d\vec{x}^2\right], \tag{9}$$

where $t \in \mathbb{R}$, $r > 0$, and $\vec{x} \in \mathbb{R}^{d-1}$. For $\phi$, we consider a radial ansatz: $\phi = \phi(r)$. The dual scalar operator $\mathcal{O}$ is then a constant boundary deformation. By the AdS/CFT dictionary [56], its conformal dimension $\Delta$ satisfies a mass-dimension relation,

$$m^2 = \Delta(\Delta - d). \tag{10}$$

Plugging this into the Klein-Gordon equation and combining the result with the $tt$ and $rr$ components of the Einstein + scalar equations yields a set of ODEs,

$$\phi'' + \left(\frac{f'}{f} - \frac{d-1}{r} - \frac{\chi'}{2}\right)\phi' + \frac{\Delta(d-\Delta)}{r^2 f}\phi = 0, \tag{11}$$

$$\chi' - \frac{2f'}{f} - \frac{\Delta(d-\Delta)\phi^2}{(d-1)rf} - \frac{2d}{rf} + \frac{2d}{r} = 0, \tag{12}$$

$$\chi' - \frac{r}{d-1}\left(\phi'\right)^2 = 0, \tag{13}$$

in agreement with [24] when $d = 3$ and $m^2 = -2$.

We are concerned with black holes, so we suppose that $f$ has a simple root at $r = r_+$—this is the horizon. Furthermore while the conformal boundary is located at $r = 0$, the singularity in our coordinates is at $r = \infty$. We must also have regularity of the metric (9) at the horizon. To emphasize this point, we note the existence of infalling coordinates in which the metric takes the form,

$$ds^2 = \frac{1}{r^2}\left[-f(r)e^{-\chi(r)}du^2 + 2e^{-\chi(r)/2}du\,dr + d\vec{x}^2\right]. \tag{14}$$

Lastly observe that the above ansatz with a particular choice of the radial functions becomes the AdS-Schwarzschild black hole—take $f(r) = 1 - (r/r_+)^d$ and $\chi(r) = 0$. The equations of motion (specifically (12)) then imply,

$$\phi(r) = 0, \tag{15}$$

so AdS-Schwarzschild is indeed the vacuum solution with no backreaction from $\phi$.

We now describe the asymptotic behavior of the radial functions as well as the corresponding field theoretic data both near the UV boundary theory ($r \to 0$) and near the IR singularity theory ($r \to \infty$). Additionally we discuss how the bulk represents an RG flow from one to the other, allowing us to treat the near-singularity data as emergent from the near-boundary data.

## 2.1 Near-Boundary Expressions and Data

The near-boundary ($r \to 0$) expressions are standard to AdS/CFT. While there is some subtlety upon which we expand in Appendix A, the key point is that a relevant scalar operator with conformal dimension $\Delta < d$ is precisely dual to a bulk scalar field with negative $m^2$ but satisfying the *Breitenlohner-Freedman stability bound* [57],

$$-\frac{d^2}{4} \le m^2 < 0. \tag{16}$$

However, we restrict to operator dimension above the unitarity bound,

$$\Delta \ge \frac{d-2}{2}. \tag{17}$$

Thus for any value of $m^2$ between $-d^2/4$ and $1-d^2/4$, the mass-dimension relation (10) gives us two possibilities for $\Delta$ depending on the boundary conditions of $\phi$ [58, 59].[8] [24] uses the "canonical" quantization in which the larger candidate (which is more strictly bounded from below by $d/2$) is chosen, but we can still take any $\Delta$ above the unitarity bound. While the bulk equations of motion are only sensitive to $m^2$, this choice will affect our interpretations of the leading-order and next-to-leading-order modes in the near-boundary expressions.

With that in mind, we now write the $\Delta \ne d/2$ near-boundary mode expansion of the field $\phi(r)$ in terms of the boundary *source* $\phi_0$ and the *one-point function* $\langle \mathcal{O} \rangle$,

$$\phi(r) \sim \phi_0 r^{d-\Delta} + \frac{\langle \mathcal{O} \rangle}{2\Delta - d}r^{\Delta}. \tag{18}$$

Next, we use the fall-off of the $tt$ component of the metric to write,

$$-r^2 g_{tt} = f(r)e^{-\chi(r)} \sim 1 - \langle T_{tt} \rangle r^d, \tag{19}$$

---

[8]For the edge case $m^2 = -d^2/4$, we only consider the Dirichlet boundary condition to avoid a double root in the CFT two-point function when taking $\Delta = d/2$.

where $\langle T_{tt} \rangle$ is the *energy density* of the thermal state. These expressions are enough to write the near-boundary expansion of $\chi(r)$ by plugging into the equation of motion (13) and noting that $\chi(0) = 0$,

$$\chi(r) \sim \frac{d-\Delta}{2(d-1)} \phi_0^2 r^{2(d-\Delta)} + \frac{2\Delta(d-\Delta)}{d(d-1)(2\Delta-d)} \phi_0 \langle \mathcal{O} \rangle r^d$$
$$+ \frac{\Delta}{2(d-1)(2\Delta-d)^2} \langle \mathcal{O} \rangle^2 r^{2\Delta} . \tag{20}$$

For $\Delta = d/2$ however, while we still have (19), we now write the Dirichlet expression [59],

$$\phi(r) \sim \phi_0 r^{d/2} \log r , \tag{21}$$

and integrating the differential equation (13) yields,

$$\chi(r) \sim \frac{\phi_0^2}{4d(d-1)} r^d \left[ 2 + 2d \log r + d^2 (\log r)^2 \right] . \tag{22}$$

There is one more aspect of the boundary state left—the temperature $T$. Because (9) is time-independent, we may easily compute the surface gravity $\kappa$ to get $T$. Defining $\vec{K} = \partial_t$, $f'_+ = f'(r_+)$ (nonzero by assumption), and $\chi_+ = \chi(r_+)$,

$$T = \frac{\kappa}{2\pi} = \frac{1}{2\pi} \sqrt{-\frac{1}{2} (\nabla^\alpha K^\beta)(\nabla_\alpha K_\beta)} \bigg|_{r=r_+} = \frac{|f'_+| e^{-\chi_+/2}}{4\pi} . \tag{23}$$

Imposing regularity on the radial functions at the horizon, the near-boundary data—$\langle T_{tt} \rangle$ and $\langle \mathcal{O} \rangle$ in particular—can be written in terms of the dimensionless ratio $\phi_0/T^{d-\Delta}$—the *deformation parameter*. As this data determines the rest of the bulk geometry by the equations of motion, the entire holographic RG flow (for each $d$ and $\Delta$ and including the near-singularity data in the IR) is labeled by $\phi_0/T^{d-\Delta}$.

## 2.2 Near-Singularity Expressions and Data

We now write the near-singularity ($r \to \infty$) expressions for the radial functions. The scalar field is dominated by a logarithmic divergence [60,61] which we write as,

$$\phi(r) \sim (d-1)c \log r . \tag{24}$$

Here $c$ is a constant with $c = 0$ corresponding to the Schwarzschild solution to the Einstein + scalar theory. Plugging this into (13) yields the near-singularity behavior of $\chi$,

$$\chi(r) \sim (d-1)c^2 \log r + \chi_1 , \tag{25}$$

where $\chi_1$ is a constant. We then use the remaining equations of motion to write $f$, bearing in mind that it is negative in the interior.

$$f(r) \sim -f_1 r^\rho , \quad \rho = d + \left( \frac{d-1}{2} \right) c^2 . \tag{26}$$

Note that these expressions all coincide with [24] at $d = 3$. Plugging these into the metric (9) and reparameterizing the now-timelike radial coordinate as,

$$r = \tau^{-2/\rho} , \tag{27}$$

| | Near-boundary ($r \to 0$) | Near-singularity ($r \to \infty$) |
|---|---|---|
| $\phi(r)$ <br> ($\Delta \neq d/2$) | $\phi_0 r^{d-\Delta} + \dfrac{\langle \mathcal{O} \rangle}{2\Delta - d} r^\Delta$ | $\sqrt{2(d-1)(\rho-d)} \log r$ |
| $\phi(r)$ <br> ($\Delta = d/2$) | $\phi_0 r^{d/2} \log r$ | $\sqrt{2(d-1)(\rho-d)} \log r$ |
| $\chi(r)$ <br> ($\Delta \neq d/2$) | $\dfrac{d-\Delta}{2(d-1)} \phi_0^2 r^{2(d-\Delta)} + \dfrac{2\Delta(d-\Delta)}{d(d-1)(2\Delta-d)} \phi_0 \langle \mathcal{O} \rangle r^d$ <br> $+ \dfrac{\Delta}{2(d-1)(2\Delta-d)^2} \langle \mathcal{O} \rangle^2 r^{2\Delta}$ | $2(\rho-d) \log r + \chi_1$ |
| $\chi(r)$ <br> ($\Delta = d/2$) | $\dfrac{\phi_0^2}{4d(d-1)} r^d \left[ 2 + 2d \log r + d^2 (\log r)^2 \right]$ | $2(\rho-d) \log r + \chi_1$ |
| $f(r)$ | $e^{\chi(r)} \left( 1 - \langle T_{tt} \rangle r^d \right)$ | $-f_1 r^\rho$ |

Table 1: The near-boundary and near-singularity expressions for the radial functions characterizing a Kasner flow ansatz for the Einstein + scalar theory. The near-boundary data is controlled by the deformation parameter $\phi_0/T^{d-\Delta}$ while all of the near-singularity data is determined by the Kasner exponent $p_t$.

yields (up to rescalings of the spacelike coordinates and an overall factor) the Kasner universe and corresponding scalar field (3),

$$ds^2 \sim -d\tau^2 + \tau^{2p_t} dt^2 + \tau^{2p_x} d\vec{x}^2, \quad \phi(\tau) \sim -\sqrt{2} p_\phi \log \tau, \tag{28}$$

with the Kasner exponents being (in terms of $\rho$),

$$p_t = 1 - \frac{2(d-1)}{\rho}, \quad p_x = \frac{2}{\rho}, \quad p_\phi = \frac{2\sqrt{(d-1)(\rho-d)}}{\rho}. \tag{29}$$

These indeed satisfy the Kasner constraints (4) and (5). Furthermore at the Schwarzschild value $c = 0 \implies \rho = d$, we get,

$$\text{Schwarzschild:} \quad p_t = -1 + \frac{2}{d}, \quad p_x = \frac{2}{d}, \quad p_\phi = 0. \tag{30}$$

## 2.3 Emergent Kasner Exponents from Flow

We summarize the results of Sections 2.1 and 2.2 in Table 1. While these results seem disconnected, we now discuss how these different asymptotic limits are linked by the bulk RG flow.

From studying the near-boundary and near-singularity data, we have that, independently of the holographic RG flow, the former is fixed by $\phi_0/T^{d-\Delta}$ while the latter is fixed by any one of the Kasner exponents—we use $p_t$ as done in [24, 55]. However, when equipped with both the near-boundary data and the equations of motion, the entire flow is already fixed and thus sets a value for $p_t$. Thus the Kasner exponents are emergent from the deformation parameter and the flow.

The most concrete way to examine this relationship is to plot the value of $p_t$ as a function of $\phi_0/T^{d-\Delta}$. For the Einstein + scalar theory, we do so for various dimensions in Figure 3. These plots are numerically determined, requiring $d$ and $\Delta$ as inputs. The details of our methodology are discussed in Appendix B.

While the numerical values depend on $d$—even the exactly-known Schwazrschild value for $p_t$ depends on dimension—the qualitative behavior of the emergent $p_t$ appears to satisfy an

inversion-like symmetry. Specifically there is a value of $\phi_0/T^{d-\Delta}$ for which $p_t$ is maximized, whereas $\phi_0/T^{d-\Delta} \to \infty$ appears to give the Schwarzschild value.[9] However there is still a problem of fine-tuning—it is unclear how the maximum is actually set. As a preliminary observation, the numerics indicate that the corresponding $\Delta = \Delta_+$ and $\Delta = \Delta_-$ (same $m^2$) flows attain the same maximum $p_t$.

The work done in a self-interacting scalar theory [55] finds the qualitative behavior to hold more generally, but the fine-tuning problem persists—the value and location of the maximum depends on the details of the theory. How these flows may exhibit universal behavior is an interesting open question which we intend to revisit.

## 3 Page Curves as Probes of Flows

Taking (9), we add a *Karch-Randall (KR) brane* and excise space beyond it. We then study the entanglement surfaces in the resulting geometry. Before discussing the details however, we briefly review the set-up (also discussed in the Introduction).

We consider the action (6) with an additional term (with $16\pi G_{d+1} = 1$),

$$I_{RS} = 2 \int_{\mathcal{Q}} d^d x \sqrt{-h}(K - T_{RS}). \tag{31}$$

$\mathcal{Q}$ is a *Randall-Sundrum (RS) brane* with tension $T_{RS}$, induced metric $h_{ab}$, and extrinsic curvature $K_{ab}$. In RSII models characterized by end-of-the-world branes, $\mathcal{Q}$ is the boundary of a manifold solving (6) and satisfying a Neumann condition,

$$K_{ab} = (K - T_{RS})h_{ab}. \tag{32}$$

With no scalar field, $\mathcal{Q}$ is a KR brane specifically when the induced geometry is AdS$_d$. This occurs precisely when the tension is *subcritical*, satisfying the bound,

$$|T_{RS}| < d - 1. \tag{33}$$

In practice, we find such KR branes by taking some foliation of AdS$_{d+1}$ into AdS$_d$ slices, then computing their respective tensions [36]. However in our work, we are considering (scalar) backreacted geometries, so the induced geometry on the brane is not a vacuum solution. Nonetheless we still have a KR brane so long as we find a boundary for which (32) is satisfied.

---

[9]This is also stated by [24, 55], but it is not immediate from the plots. Rather, this is assumed asymptotic behavior based on running the numerics to large $\phi_0/T^{d-\Delta}$.

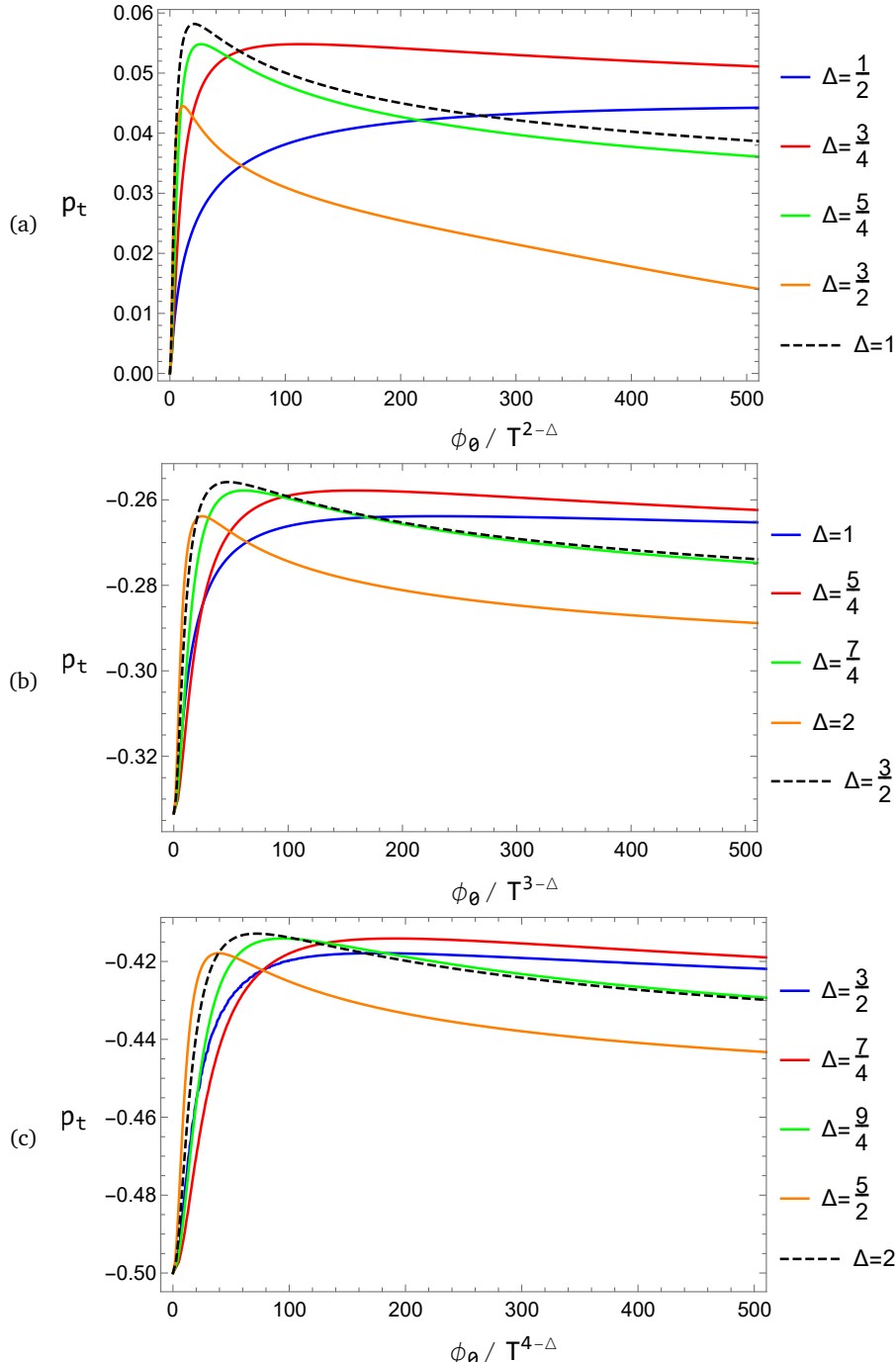

Figure 3: The emergent Kasner exponent $p_t$ versus the dimensionless deformation parameter $\phi_0/T^{d-\Delta}$ for various $d$ and $\Delta$, computed via a numerical shooting method outlined in Appendix B. We show the results for (a) $d=2$, (b) $d=3$, and (c) $d=4$. The dashed lines correspond to $\Delta = d/2$.

Einstein gravity in the bulk "localizes" to "induced" quantum gravity on the brane [11, 12, 14], although what localization means is a matter of interpretation. Agnostically, the original Karch-Randall story [11] considers the transverse-traceless (TT) modes of linearized fluctuations arising from the backreaction of the brane. Generically, these TT modes satisfy the Schrödinger equation with a "volcano" potential containing a weighted $\delta$-function term centered at the brane (see Figure 4). However, the weight of the $\delta$-function smoothly vanishes as $T_{RS} \to 0$ (i.e. the probe limit), resulting in the loss of the volcano's "crater." Regardless of the tension however, solving for the KK tower of spin-2 modes becomes a quantum mechanics problem—computing the eigenstates (labeled by their masses) of the Schrödinger equation with the volcano potential.

In the near-critical-tension regime when (33) is nearly saturated, the lowest-mass ("almost-zero") mode has a positive mass which is nonetheless much smaller than the masses in the rest of the tower [11, 62, 63]. Furthermore, the almost-zero mode's wavefunction is sharply peaked at the brane. Thus we think of the almost-zero KK mode as a graviton that is "localized" to the brane in the sense of its wavefunction dying off quickly in the bulk. The higher modes meanwhile dualize to a CFT on the brane. This gives description (II) in the Introduction for the near-critical branes.

However, we remark that there is not necessarily a problem with maintaining this interpretation as we leave the critical limit. As we tune the tension down by $O(1)$ factors, the mass spacing between the almost-zero mode and the first excited state will decrease, and the former's corresponding wavefunction will broaden. In other words, the almost-zero mode no longer has an obvious interpretation as a localized graviton. Nonetheless, these changes are quantitative rather than qualitative, and there is no sign of a discontinuity in the bulk analysis. Thus, one may advocate (as we do so here) that viewing the almost-zero mode as a graviton coupled to a CFT obtained from the higher modes is still a valid holographic interpretation of the picture.[10] Indeed, [13] considers such a scenario at a tension which is certainly subcritical.[11] However, the theory of gravity here would be far more unusual than in the near-critical limit in that there is stronger mixing between the graviton and the CFT, since the states of the KK tower are all "closer together."

The fine-tuning which pushes the limits of this reasoning the most is the tensionless case, in which we lack both the almost-zero-mode wavefunction's sharp peak and, as stated above and shown in Figure 4, the crater of the volcano potential. Both features might be viewed as signals of "localization." However, there is still no discontinuity when going from nonzero to zero tension; indeed, the crater vanishes smoothly as seen in the volcano potential written in [11] for $d = 4$. Along the reasoning of [27] (which, as we do here, studies entanglement islands), the tensionless braneworld theory is then "no worse" than that of nonzero tension, at least away from the near-critical regime.

With dynamical gravity on the brane, the conformal boundary acts as a "bath" into which information may flow from the brane theory. We thus have a natural arena in which to explore black hole information—simply put a black hole on the brane itself. Furthermore as we have a scalar deformation on the bath controlled by $\phi_0/T^{d-\Delta}$, we can observe how tuning the deformation affects information.

To further take a semiclassical approximation, we consider the leading-order effective induced action—identified as $d$-dimensional Einstein gravity from the series worked out by [14]. The induced Einstein action is dynamical for $d > 2$ but not for $d = 2$ (for which having a dynamical effective theory would require inclusion of the next-to-leading order Polyakov term $\sim R \log |R|$). We thus only consider $d > 2$ and leave explorations of the relationship between scalar deformations and Page curves in $d = 2$ to future work.

---

[10]We thank Andreas Karch for clarifying this point and its corresponding rationale to us.

[11] [13] considers $d = 4$. The critical tension is 3, but they use a brane with tension $3/\sqrt{2}$.

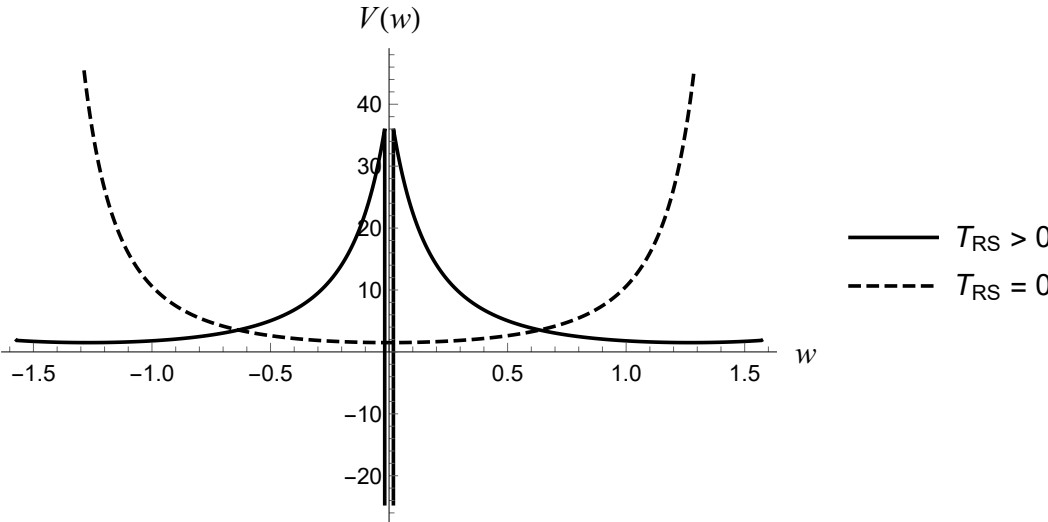

Figure 4: A sketch of the AdS volcano potential $V(w)$ in [11] for $d = 4$, depicted as a function of a coordinate $w$ which is normal to the brane. We show both the $T > 0$ case (which has a crater) and the $T = 0$ case (which lacks a crater).

Concretely, we compute holographic entanglement entropy. We assume that the appropriate semiclassical holographic prescription is still the island rule [2–5] but in the backreacted geometry—for some radiation region $\mathcal{R}$ on the boundary as shown in Figure 1, its entanglement entropy to leading order as $G_{d+1} \to 0$ is given as,

$$S(\mathcal{R}) = \frac{A(\gamma)}{4G_{d+1}} \,. \tag{34}$$

Here $\gamma$ is the minimal-area surface in the island rule, so it can either be entirely homologous to $\mathcal{R}$ or be homologous to $\mathcal{R} \cup \mathcal{I}$ where $\mathcal{I}$ is an entanglement island residing on the brane. We specifically study two-sided black holes, so the radiation region consists of a left piece $\mathcal{R}^L$ and right piece $\mathcal{R}^R$ as in Figure 2.

Using (34) however requires fixing a particular KR brane and thus setting a tension $T_{RS}$. For mathematical simplicity, we consider the probe limit $T_{RS} = 0$; islands with nonzero tension have been studied by [13–15,17,27] but we leave how this part of the story interacts with scalar deformations of the bath open to exploration. We also reiterate that, although the holographic interpretation of the tensionless "brane + bath" theory is highly nonstandard gravity due to the lack of separation between the almost-zero and excited spin-2 KK modes, in following [27] we proceed on the basis of this scenario being not particularly worse than those of nonzero-tension braneworlds.

Regarding the position of the brane, picking-out a transverse coordinate $x^1$ in $\vec{x} = (x^1, ..., x^{d-1})$ as shown in the metric (9), we take the slice,

$$x^1 = 0 \,. \tag{35}$$

We must ensure that this is a proper KR brane—that given the full action (6) plus the RS term, both the geometry (9) and the radial scalar field ansatz $\phi = \phi(r)$ discussed in Section 2 satisfy boundary conditions on $\mathcal{Q}$. Starting with the former, we note that the normal unit vector is,

$$n_\mu = \frac{1}{r}\delta_{\mu 1} \,, \tag{36}$$

where $\delta_{\mu 1}$ is the Kronecker delta and 1 denotes the $x^1$ coordinate. The resulting extrinsic curvature on the (35) slice is then,

$$K_{ab} = 0 \,, \tag{37}$$

so this is certainly a tensionless KR brane.

As for the scalar field $\phi = \phi(r)$, it clearly satisfies a Neumann boundary condition because it is independent of $x^1$,

$$n^\mu \partial_\mu \phi(r) = r \partial_1 \phi(r) = 0 \,. \tag{38}$$

The punchline is that, for a particular scalar deformation—that is some deformation parameter $\phi_0/T^{d-\Delta}$—there exist radiation regions for which the entanglement entropy between the left and right parts of a two-sided black hole with a KR brane obeys a Page curve (Figure 2). There are technically three parameters which can be tuned given $d$: $\phi_0/T^{d-\Delta}$, the operator dimension $\Delta$, and the endpoint of the radiation region $x_\mathcal{R}$ (on both sides—we take $\mathcal{R}^L$ and $\mathcal{R}^R$ to start at $x^1 = x_\mathcal{R}$).

The primary parameter of interest is $\phi_0/T^{d-\Delta}$ because, by keeping $\Delta$ and $x_\mathcal{R}$ fixed, the resulting Page curves correspond directly to bulk RG flows from the same equations of motion. We also study how changing $\Delta$ affects the Page curve. The underlying motivation behind this analysis is to understand how changing the bath deformation may alter the physics, so we leave $x_\mathcal{R}$ alone (although we need to justify that such a radiation region which can be kept stable despite tuning the deformation indeed exists—see Section 3.1 for details).

Nonetheless, it is also interesting to understand what happens when we tune $x_\mathcal{R}$. To do so meaningfully, we fix $d = 3$ and $\Delta = 2$, comparing the AdS-Schwarzschild solution ($\phi_0/T = 0$) to a solution for a nontrivial deformation. We find that larger $x_\mathcal{R}$ correspond to more direct probes of backreaction in the interior specifically.

## 3.1 The Page Point

When tuning the scalar deformation, a natural question arises: can we definitively choose a radiation region $\mathcal{R}$ which yields a Page curve regardless of the values of $\phi_0/T^{d-\Delta}$ and $\Delta$? We thus define the *Page point* $x_p$ as the value of $x_\mathcal{R}$ for which,

$$0 \leq x_\mathcal{R} \leq x_p \implies \text{no Page curve for } S(\mathcal{R}) \,, \tag{39}$$

$$x_\mathcal{R} > x_p \implies \text{Page curve for } S(\mathcal{R}) \,. \tag{40}$$

The analysis below essentially follows [17] but for a probe brane (35), general blackening factor $f(r)$, and various dimensions. [17] confirms the existence of Page points in AdS-Schwarzschild for $d = 4$.

Specifically, we restrict our attention to the $t = 0$ slice on which the interior is trivial. Given some radiation region, we will ultimately find two candidates for the entanglement surface (Figure 5): Hartman-Maldacena surfaces [34] which span the black hole without hitting the brane and *island surfaces* which hit the brane outside of the horizon [19]. The Hartman-Maldacena surfaces grow away from the $t = 0$ slice because of the growth in the Einstein-Rosen bridge, whereas the island surfaces are time-independent and thus maintain a constant area.

For a particular radiation region defined by $x_\mathcal{R}$, we can identify which of these surfaces is "initially" minimal and deduce whether or not we get a Page curve—this happens if the Hartman-Maldacena surface is minimal at $t = 0$.[12] We will ultimately compute the Page point for various dimensions as a function of $\phi_0/T^{d-\Delta}$.

Jumping into the calculation, the geometry of the $t = 0$ slice is,

$$ds^2|_{t=0} = \frac{1}{r^2}\left[\frac{dr^2}{f(r)} + d\vec{x}^2\right] = \frac{1}{r^2}\left[\frac{dr^2}{f(r)} + \sum_{i=1}^{d-1}(dx^i)^2\right]. \tag{41}$$

---

[12]Note that in our calculations, we will be computing the portions of the area in just one of the exterior patches and multiplying by 2 to get the $t = 0$ areas.

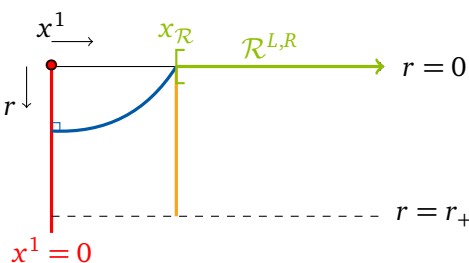

Figure 5: The candidate RT surfaces in one of the exterior regions. The orange line entering the horizon is the Hartman-Maldacena surface, whereas the blue arc which ends perpendicularly on the brane is the island-producing exterior surface.

Parameterizing an arbitrary surface as $x^1 = x^1(r)$, its area functional is,

$$A = \int dx^2 \cdots dx^{d-1} \int \frac{dr}{r^{d-1}} \sqrt{\frac{1}{f(r)} + \dot{x}^1(r)^2}, \quad \dot{x}^1 = \frac{dx^1}{dr}. \tag{42}$$

Integrating over $(x^2, ..., x^{d-1})$ produces an overall volume factor $V_{d-2}$ which is irrelevant when comparing different surfaces $x^1(r)$ together, so we only care about the *area density* (also called "area" for convenience) $\mathcal{A}$ and its Lagrangian $\mathcal{L}$,

$$\mathcal{A} = \frac{A}{V_{d-2}} = \int \frac{dr}{r^{d-1}} \sqrt{\frac{1}{f(r)} + \dot{x}^1(r)^2} = \int dr \, \mathcal{L}. \tag{43}$$

We want to minimize $\mathcal{A}$. The Euler-Lagrange equations indicate that $\partial \mathcal{L}/\partial \dot{x}^1$ is a constant of motion. In fact, defining the *turnaround point* $r = r_T > 0$ as the root of $1/\dot{x}^1(r)$ (so $dr/dx^1 = 0$ here), we have that minimal surfaces satisfy,

$$\frac{\dot{x}^1(r)}{r^{d-1}\sqrt{\frac{1}{f(r)} + \dot{x}^1(r)^2}} = \pm \frac{1}{r_T^{d-1}} \implies \dot{x}^1(r) = \pm \frac{r^{d-1}}{\sqrt{f(r)\left[r_T^{2(d-1)} - r^{2(d-1)}\right]}}. \tag{44}$$

These surfaces must also satisfy boundary conditions at the conformal boundary ($r = 0$) and at the brane ($x^1 = 0$). Because of the branching in (44), the conditions must be derived by assuming a more general parameterization $(r(s), x^1(s))$ with $s \in [0, 1]$ [27]. They are respectively a Dirichlet condition and a Neumann condition,

$$\text{Boundary: } x^1(0) = x_{\mathcal{R}}, \tag{45}$$

$$\text{Brane: } \left.\frac{1}{\dot{x}^1(r)}\right|_{x^1=0} = 0. \tag{46}$$

So for some $x_{\mathcal{R}}$, the minimal surfaces in the presence of a probe brane are those which satisfy (44) and either do not end on the brane (Hartman-Maldacena),

$$r_T = \infty \implies x^1(r) = x_{\mathcal{R}}, \tag{47}$$

or do end on the brane (island surfaces). For the latter case, the turnaround point is precisely along the brane, i.e.,[13]

$$x^1(r_T) = 0, \tag{48}$$

---

[13]These statements come from viewing the tensionless braneworld as orbifolded AdS$_{d+1}$ [17,35].

but the above equations of motion constrain the value of $r_T$ to depend on $x_{\mathcal{R}}$. Specifically by integrating (44) (noting that the only branch which reaches the brane from $x_{\mathcal{R}} > 0$ is the negative one), we have,

$$\int_0^{r_T} dr \frac{r^{d-1}}{\sqrt{f(r)\left[r_T^{2(d-1)} - r^{2(d-1)}\right]}} = x_{\mathcal{R}}. \tag{49}$$

We now use (43) to write the $t = 0$ areas of both the Hartman-Maldacena surface and the island surface given some $x_{\mathcal{R}}$. Respectively they are (taking the area in one exterior patch and multiplying by 2),

$$\mathcal{A}_{HM}(0) = 2 \int_0^{r_+} \frac{dr}{r^{d-1}\sqrt{f(r)}}, \tag{50}$$

$$\mathcal{A}_I = 2 \int_0^{r_T} \frac{dr}{r^{d-1}\sqrt{f(r)}} \frac{r_T^{d-1}}{\sqrt{r_T^{2(d-1)} - r^{2(d-1)}}}. \tag{51}$$

Both of these are divergent at the boundary, so we will need to renormalize them. We use the standard holographic renormalization—integrating from a cutoff $r/r_+ = \epsilon \ll 1$ then adding the appropriate counterterm before taking $\epsilon \to 0$. For both of these areas, the integrands go to $1/r^{d-1}$ as $r \to 0$, so the counterterm is the same as for areas in empty $\mathrm{AdS}_{d+1}$ and only depends on the dimension,

$$r_+^{d-2} \mathcal{A}_{CT} = -\frac{2}{(d-2)} \frac{1}{\epsilon^{d-2}}. \tag{52}$$

While this is important for evaluating Page curves, in determining whether or not there is a Page curve to begin with we only need to use the area difference,

$$\Delta\mathcal{A}(0) = \mathcal{A}_I - \mathcal{A}_{HM}(0)$$
$$= 2 \int_0^{r_T} \frac{dr}{r^{d-1}\sqrt{f(r)}} \left[ \frac{r_T^{d-1} - \sqrt{r_T^{2(d-1)} - r^{2(d-1)}}}{\sqrt{r_T^{2(d-1)} - r^{2(d-1)}}} \right] - 2 \int_{r_T}^{r_+} \frac{dr}{r^{d-1}\sqrt{f(r)}}, \tag{53}$$

which is UV-finite. There is then a Page curve if and only if $\Delta\mathcal{A}(0) > 0$.

If we have AdS-Schwarzschild—that is $f(r) = 1 - (r/r_+)^d$—then it is straightforward to plot $\Delta\mathcal{A}(0)$ as a function of $x_{\mathcal{R}}$ using (49) and (53). We do this for various dimension in Figure 6, thus finding the Page points $x_p$ in such geometries. Numerically, these Page points (in dimensionless coordinates) for $d = 3, 4$ are,

$$\frac{x_p}{r_+} \approx \begin{cases} 0.589, & \text{if } d = 3, \\ 0.444, & \text{if } d = 4. \end{cases} \tag{54}$$

We now extend this analysis to nonzero $\phi_0/T^{d-\Delta}$—increasing the deformation parameter changes $f(r)$ and thus the calculation of $\mathcal{A}$. We again utilize a shooting method, this time numerically solving for $f(r)$ in the exterior. Doing so allows us to compute the Page points in a particular $d$ as a function of $\phi_0/T^{d-\Delta}$ (Figure 7).

Observe that the Page point decreases as we increase $\phi_0/T^{d-\Delta}$—a large scalar deformation results in Page curves for larger radiation regions. Notably, if we take a radiation region $x_{\mathcal{R}} > x_p$ with zero deformation (i.e. in AdS-Schwarzschild), then we may tune $\phi_0/T^{d-\Delta}$ up while keeping $x_{\mathcal{R}}$ fixed without losing the Page curve. This justifies the viewpoint of our work—that the Page curves for a particular radiation region probe Kasner flows.

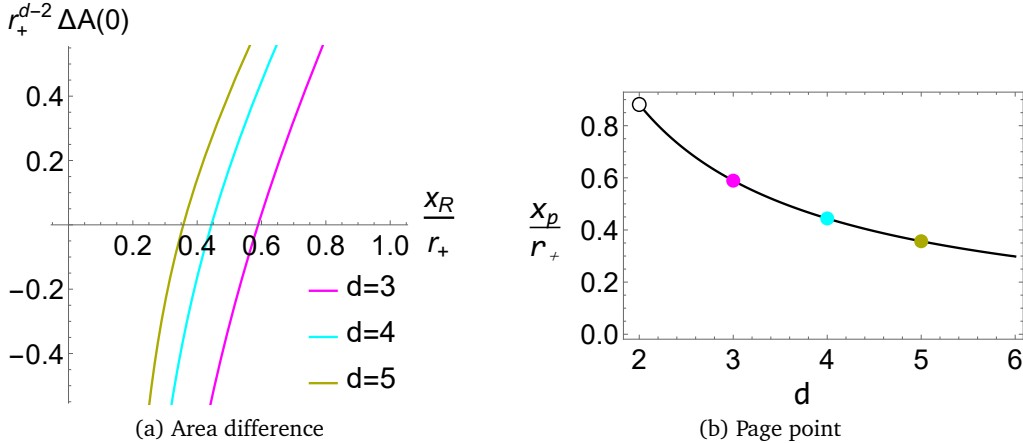

(a) Area difference                    (b) Page point

Figure 6: (a) The area difference $\Delta\mathcal{A}(0)$ (53) versus $x_{\mathcal{R}}$ (49) for $d = 3, 4, 5$ and (b) the Page point $x_p$ as a function of (analytically-continued) $d > 2$, all in the AdS-Schwarzschild geometry and in $r_+$ units. The area difference in each dimension monotonically increases with $x_{\mathcal{R}}$, crossing 0 at a particular value beyond which we have Page curves. The Page point thus approaches the brane as we increase $d$.

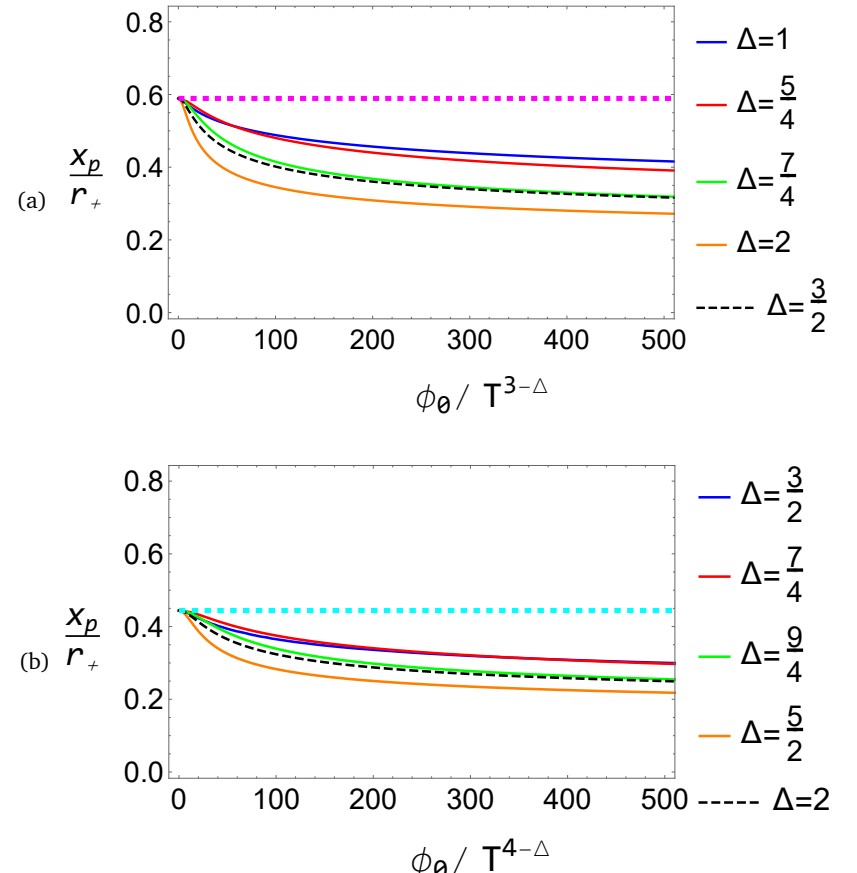

Figure 7: The Page point $x_p$ as a function of $\phi_0/T^{d-\Delta}$ for (a) $d = 3$ and (b) $d = 4$ (the cases in Section 3.2). The horizontal lines are at the AdS-Schwarzschild Page points shown in Figure 6. The Page point decreases as $\phi_0/T^{d-\Delta}$ increases.

## 3.2 Entanglement Entropy and Page Curves

We now compute the Page curves for various dimensions $d$ and conformal dimensions $\Delta$. To reiterate, the endpoint $x_{\mathcal{R}}$ of the radiation region is also a tunable parameter, but in studying the effect of changing the scalar deformation we start by keeping it fixed. For much of the analysis below, we will assume,

$$x_{\mathcal{R}} = 2x_p \,, \tag{55}$$

which ensures that we always have a Page curve with finite Page time. We subsequently briefly touch on the physics of increasing $x_{\mathcal{R}}$.

However, while the island surfaces are constant in time, the Hartman-Maldacena surfaces are not—we merely know them to be constant-$x^1$ surfaces. In Section 3.1 we computed the areas of these surfaces at $t = 0$, so we now consider the areas as functions of time. Only then can we plot the Page curves themselves.

### 3.2.1 Growth of the Hartman-Maldacena Surfaces

The procedure for computing the time-dependent Hartman-Maldacena surfaces is outlined by [34]. Since the analysis in this particular section is blind to KR branes, we include $d = 2$ here. For a constant-$x^1$ slice,[14] the induced metric from (9) is,

$$ds^2|_{x^1 = x_{\mathcal{R}}} = \frac{1}{r^2}\left[ -f(r)e^{-\chi(r)}dt^2 + \frac{dr^2}{f(r)} + \sum_{i=2}^{d-1}(dx^i)^2 \right]. \tag{56}$$

For a surface $r = r(t)$, the area functional is then,

$$A = V_{d-2} \int \frac{dt}{r(t)^{d-1}} \sqrt{-f[r(t)]e^{-\chi[r(t)]} + \frac{\dot{r}(t)^2}{f[r(t)]}}, \quad \dot{r} = \frac{dr}{dt}, \tag{57}$$

with the area (density) functional and corresponding Lagrangian being,

$$\mathcal{A} = \int \frac{dt}{r(t)^{d-1}} \sqrt{-f[r(t)]e^{-\chi[r(t)]} + \frac{\dot{r}(t)^2}{f[r(t)]}} = \int dt \, \mathcal{L}. \tag{58}$$

The lack of explicit $t$-dependence in this Lagrangian gives us a constant of motion which we identify as the "energy" $E$ of the minimal surface (suppressing $t$),

$$E = \dot{r}\frac{\partial \mathcal{L}}{\partial \dot{r}} - \mathcal{L} = \frac{f(r)e^{-\chi(r)}}{r^{d-1}\sqrt{-f(r)e^{-\chi(r)} + \frac{\dot{r}^2}{f(r)}}}, \tag{59}$$

so the minimal trajectories are defined by,

$$\dot{r} = \pm f(r)e^{-\chi(r)/2}\sqrt{1 + \frac{f(r)e^{-\chi(r)}}{(r^{d-1}E)^2}}. \tag{60}$$

We may conventionally take the sign of $E$ to match the sign of (60). Now observe that $\dot{r} = 0$ in the interior (when $f < 0$) if there is a radius $r = r_*$ such that,

$$-\frac{f(r_*)e^{-\chi(r_*)}}{r_*^{2(d-1)}} = E^2 \,. \tag{61}$$

---

[14]The analysis below is blind to which constant-$x^1$ slice we take.

So for a surface with such an energy $E$, this is the maximal value of $r$ assumed. In full, the surface starts at the conformal boundary, reaches $r = r_*$, then goes to the other side of the black hole. Thus (reparameterizing as $t = t(r)$ and using $dt/dr = 1/\dot{r}$), we have that the area is,

$$\mathcal{A}_{HM}(t_b) = 2 \int_0^{r_*} \frac{dr}{r^{d-1}\sqrt{f(r) + e^{\chi(r)}(r^{d-1}E)^2}} \implies \dot{r}|_{r=r_*} = 0. \tag{62}$$

Here, we refer to the Hartman-Maldacena area as being a function of the *boundary time* $t_b$. This is to avoid confusion with the bulk time coordinate.[15] We find it by considering the integral,

$$\int_0^{r_*} \frac{dr}{\dot{r}} = t_* - t_b, \tag{63}$$

which computes the time difference between $t_b$ and $t_* = t(r_*)$. By restricting our attention to *symmetric* surfaces—those for which $t_*$ is purely imaginary—we get the boundary time in terms of $E$ [24] (removing the pole at the horizon),

$$t_b = -P \int_0^{r_*} \frac{\text{sgn}(E)e^{\chi/2}}{f(r)\sqrt{1 + f(r)e^{-\chi(r)}/(r^{d-1}E)^2}}. \tag{64}$$

To summarize, $E$ is a parameter for the Hartman-Maldacena surfaces which determines the maximal radius $r_*$, the boundary time $t_b$, and the area $\mathcal{A}_{HM}$. We restrict consideration to $E \geq 0$—this keeps us in the $t_b \geq 0$ regime.

To conclude this discussion, we analyze the late-time ($t_b \to \infty$) behavior of these surfaces. It is first necessary to observe that the function,

$$g(r) = -\frac{f(r)e^{-\chi(r)}}{r^{2(d-1)}}, \tag{65}$$

has a maximum at some *critical radius* $r = r_c$ in the interior. This is because,[16]

$$g(r_+) = 0, \quad \lim_{r\to\infty} g(r) = \lim_{r\to\infty} f_1 e^{-\chi_1} r^{-(\rho-2)} = 0, \quad g(r)|_{r>r_+} > 0. \tag{66}$$

We have used the asymptotic expressions for $f$ and $\chi$ (Table 1). Denoting the corresponding energy for the surface with $r_* = r_c$ by $E = E_c$, (61) indicates that,

$$1 + \frac{f(r_c)e^{-\chi(r_c)}}{(r_c^{d-1}E_c)^2} = 0, \tag{67}$$

so the integral for the boundary time diverges, i.e. $t_b \to \infty$ as $r_* \to r_c$. This surface is in a sense the "maximal" one.

Now by plugging into (58) as in [34], we can obtain the linear-growth term for $\mathcal{A}_{HM}$ and hence for the entanglement entropy $S = V_{d-2}\mathcal{A}_{HM}/(4G_{d+1})$. Specifically for late boundary times, we use the integrand to write,[17]

$$\frac{\partial \mathcal{A}_{HM}}{\partial t_b} = -2\frac{f(r_c)e^{-\chi(r_c)}}{r_c^{2(d-1)}}\frac{1}{|E_c|} = 2\sqrt{-\frac{f(r_c)e^{-\chi(r_c)}}{r_c^{2(d-1)}}} = \frac{2v}{r_+^{d-1}}, \tag{68}$$

---

[15]The specific case of $t_b = 0$ coincides exactly with the analysis on the $t = 0$ slice. We can confirm this by noting that the corresponding energy is $E = 0$.

[16]This argument breaks down for $d = 2$ in AdS-Schwarzschild—$c = 0$ and thus $\rho = 2$. Instead, the "maximum" is at infinity, so we take $r_c = \infty$. The late-time Hartman-Maldacena surfaces here are thus geodesics which probe the singularity itself.

[17]When writing the late-time linear behavior of entropy, one typically uses an *entropy density* $s$ in conjunction with the transverse volume. We find it to be $\sim 2/r_+^{d-1}$.

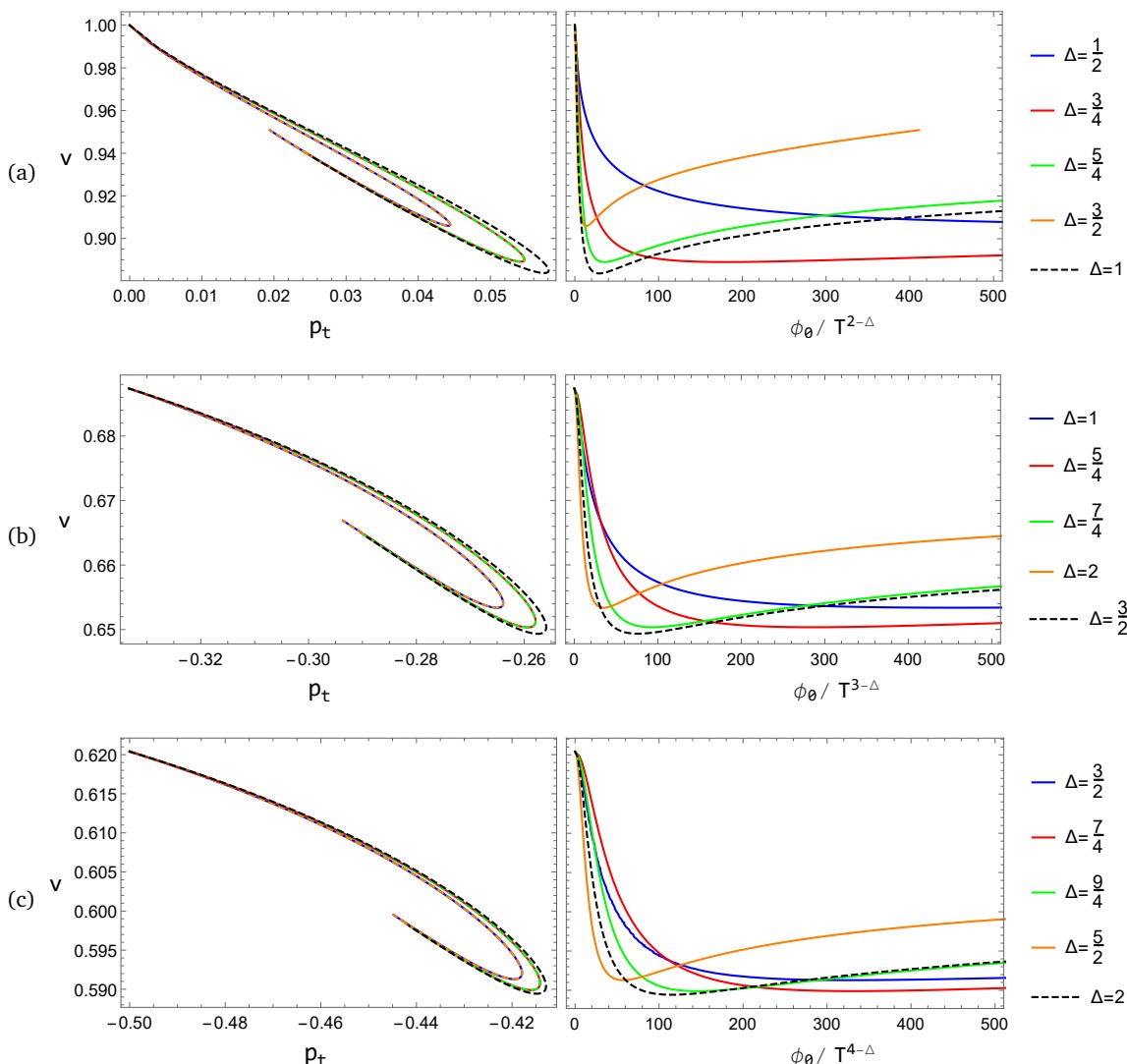

Figure 8: Entanglement velocity $v$ as a function of $p_t$ and $\phi_0/T^{d-\Delta}$ for (a) $d = 2$, (b) $d = 3$, and (c) $d = 4$. Each plot starts at its AdS-Schwarzschild value from (69). The $p_t$ curves obey a $\Delta \leftrightarrow d - \Delta$ symmetry—they are sensitive to $m^2$.

where $v$ is the entanglement velocity—a dimensionless factor which captures the speed of entropy growth—as written by [24]. In AdS-Schwarzschild [34],

$$v_{AS} = \frac{\sqrt{d}(d-2)^{(d-2)/(2d)}}{[2(d-1)]^{(d-1)/d}} . \tag{69}$$

[24] plots $v$ in the Kasner metric in terms of $p_t$ for $d = 3$, $\Delta = 2$. As we care about boundary data, we present analogous plots for various $d$ and $\Delta$ in Figure 8.

For all relevant deformations used in our numerics—including $\Delta > d/2$, $\Delta < d/2$, and $\Delta = d/2$—as a function of $\phi_0/T^{d-\Delta}$ the entanglement velocity decreases from the AdS-Schwarzschild value $v_{AS}$ until reaching a minimal value, then slowly increases back towards $v_{AS}$. This is much like what we see in the behavior of the Kasner exponent $p_t$ (Figure 3). Essentially this indicates a nontrivial relationship between the scalar deformation and the "speed" of entanglement in the bath.

Regarding the Page time, we still need to compare the Hartman-Maldacena surfaces to the island surfaces. These also change under the scalar deformation but additionally depend on $x_{\mathcal{R}}$ (whereas the Hartman-Maldacena surfaces alone do not).

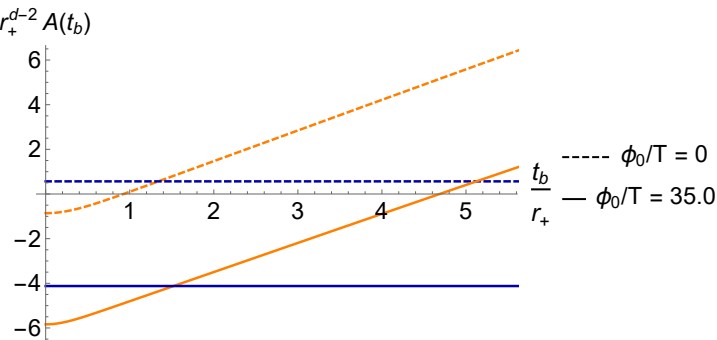

Figure 9: The Page curves for $\phi_0/T = 0$ and $\phi_0/T \approx 35.0$ with $d = 3$, $\Delta = 2$, presented in terms of the area $\mathcal{A}$ and boundary time $t_b$ in units of $r_+$. The orange curves are Hartman-Maldacena surface areas while the blue curves are island surface areas. For both, we observe a phase transition from the former to the latter at finite boundary times—the respective Page times given these deformations.

### 3.2.2 Initial Area Difference and the Page Time

We are now equipped to compute Page curves given some $d$, $\Delta$, and $x_\mathcal{R}$ (returning to the KR model and $d > 2$). As we wish to understand the physics of the scalar deformation, we first keep the radiation region fixed at $x_\mathcal{R} = 2x_p$ (55), thus ensuring a Page curve regardless of the deformation. We present our numerics in terms of the area and boundary time in units of $r_+$.

As a preliminary exercise, we numerically compute the Page curve for $d = 3$, $\Delta = 2$ for both $\phi_0/T = 0$ and $\phi_0/T \approx 35.0$ (the latter being close to the minimum for $v$). To do so, we simply use the expressions from Section 3.2.1 to acquire the entropy from the Hartman-Maldacena surface and those of Section 3.1 to get its upper bound—the entropy from the island surface. We renormalize both of the areas using the standard counterterm (52). The result is shown in Figure 9.

To confirm that our numerics are appropriately tuned, we perform two checks. The first is to compare the initial area $r_+^{d-2}\mathcal{A}_{HM}(0)$ of the $\phi_0/T = 0$ curve to the analytical result (Appendix C). Indeed for $d = 3$, we have,

$$r_+^{d-2}\mathcal{A}_{HM}(0)|_{\phi_0/T=0} = -\frac{2\sqrt{\pi}\Gamma\left(\frac{2}{3}\right)}{\Gamma\left(\frac{1}{6}\right)} \approx -0.862 \,, \tag{70}$$

which closely aligns with Figure 9.

The second check is to compare the entanglement velocities (found from halving the late-time slope of the Hartman-Maldacena surface areas (68)) against both each other and the values seen in previous sections. According to our numerics,

$$v|_{\phi_0/T=0} \approx 0.687 \,, \quad v|_{\phi_0/T\approx35.0} \approx 0.653 \,. \tag{71}$$

These match Figure 8, with $v|_{\phi_0/T=0}$ also being the AdS-Schwarzschild value found from (69).

Observe that the entanglement velocity is not the only thing which changes—the scalar deformation also affects the initial difference $\Delta\mathcal{A}(0)$ between the Hartman-Maldacena and island surfaces to $O(10^{-1})$,

$$\Delta\mathcal{A}(0)|_{\phi_0/T=0} \approx 1.43 \,, \quad \Delta\mathcal{A}(0)|_{\phi_0/T\approx35.0} \approx 1.72 \,. \tag{72}$$

Furthermore the Page times read from Figure 9 are,

$$\frac{t_p}{r_+}\bigg|_{\phi_0/T=0} \approx 1.34 \,, \quad \frac{t_p}{r_+}\bigg|_{\phi_0/T\approx35.0} \approx 1.52 \,. \tag{73}$$

Indeed, we may ask which effect is more significant in influencing the Page time: the change to the entanglement velocity or the change to the area difference. Assuming linear behavior by the Hartman-Maldacena area—a loose but apparently safe approximation within $t_b/r_+ \sim O(1)$—we can estimate the Page time as,

$$t_p \approx \frac{\Delta\mathcal{A}(0)}{2\nu}, \tag{74}$$

so the "first-order" variation in the Page time $\delta t_p$ is related to the variations in initial area difference and entanglement velocity by,

$$\delta t_p \approx \frac{\delta(\Delta\mathcal{A}(0))}{2\nu} - \frac{\Delta\mathcal{A}(0)}{2\nu^2}\delta\nu. \tag{75}$$

So for the numbers above and using the average values of $\Delta\mathcal{A}(0)$ and $\nu$, this estimation yields (writing both terms individually and setting $r_+ = 1$),

$$\delta t_p \approx 0.22 + 0.06 = 0.28. \tag{76}$$

The point is not exactness, but to demonstrate that the contribution from $\delta(\Delta\mathcal{A}(0))$ (of $O(10^{-1})$) is more significant than the contribution from $\delta\nu$ (of $O(10^{-2})$). The former comes from backreaction in the exterior while the latter comes from backreaction in the interior. As our numerics indicate that $\delta t_p$ is actually closer to $0.19 \sim O(10^{-1})$, it is the former effect—the exterior backreaction—which has a greater influence on the Page time.

To obtain a more complete picture, we numerically compute two additional sets of plots, expanding our analysis to $d = 3, 4$ and a range of $\Delta$. The first set is the initial area difference $\Delta\mathcal{A}(0)$ as a function of $\phi_0/T^{d-\Delta}$. The second is the Page time $t_p$ as a function of $\phi_0/T^{d-\Delta}$. We present our results in Figure 10.

The numerics are unstable at larger values of $\phi_0/T^{d-\Delta}$, which is why we restrict to $\sim 1500$. In spite of this limitation, there are lessons in how the plots for different $\Delta$ relate to one another. Firstly, the $\phi_0/T^{d-\Delta} = 0$ values—when the deformation is turned off—has no dependence on $\Delta$. There is no backreaction and thus, unsurprisingly, the geometry is just AdS-Schwarzschild.

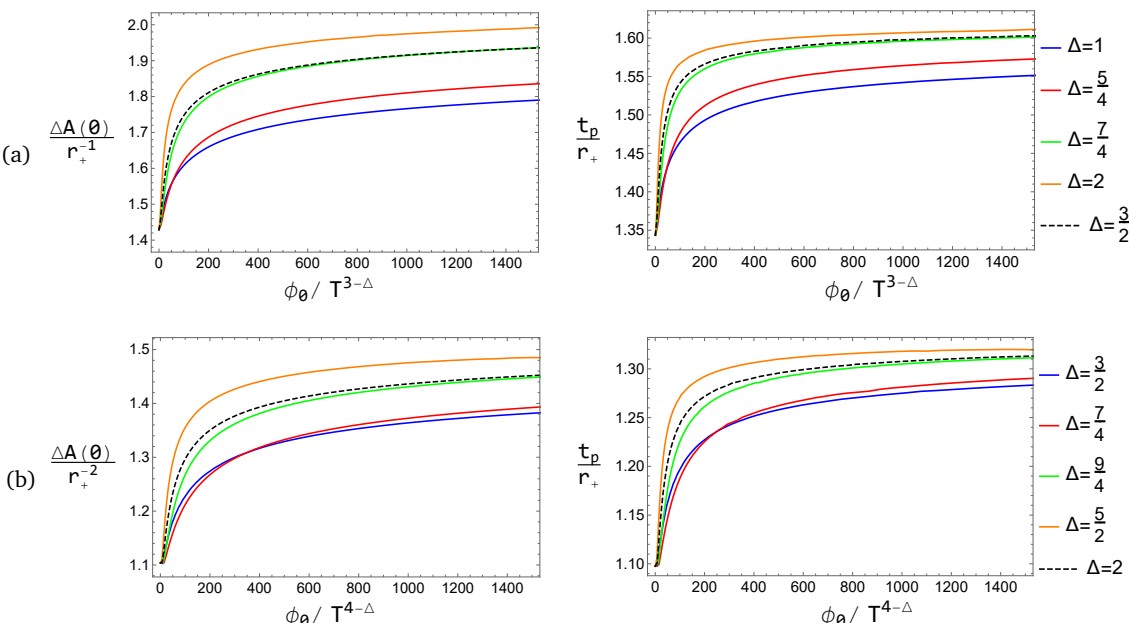

Figure 10: The initial area difference $\Delta\mathcal{A}(0)$ and the Page time $t_p$ plotted as functions of $\phi_0/T^{d-\Delta}$ for (a) $d = 3$ and (b) $d = 4$, all for various $\Delta$. The axes are in $r_+$ units.

For all $\Delta$, as we increase the deformation parameter both the initial area and the Page time grow, seemingly together with the range of variation being $O(10^{-1})$. This is evidence for the statement above: the changes to the entanglement velocity are so small that it is indeed the changes to the initial area difference which more strongly inform the changes in the Page time.

Noteworthy is the large $\phi_0/T^{d-\Delta}$ behavior. The Page time $t_p/r_+$ curves in particular appear to fall into two "families" of behavior—the $\Delta < d/2$ curves (corresponding to the "Neumann" $\Delta = \Delta_-$ quantization) and the $\Delta \geq d/2$ curves (corresponding to the "Dirichlet" $\Delta = \Delta_+$ quantization)[18] seem to separate from another. This is most evident for the Page time plot in $d = 4$, in which the curves in the same family actually approach one another.

Taking this behavior seriously, we may claim that upon introducing a large scalar deformation, the Page time is not sensitive to $\Delta$—only the quantization (as viewed by [58]) of the deformation, which we recall is related to the boundary condition of the scalar field [59].

### 3.2.3 Interior Versus Exterior Backreaction

In the previous section we present our results across a variety of $d$ and $\Delta$ but keeping $x_{\mathcal{R}}$ fixed at twice the Page point $x_p$. While we were able to probe the effect of simply tuning the deformation on the Page time, we found that the exterior backreaction influenced the quantitative results more than the interior backreaction. We thus now keep $d = 3$ and $\Delta = 2$, again comparing $\phi_0/T = 0$ to $\phi_0/T \approx 35.0$ but tuning $x_{\mathcal{R}}$.

The numbers of interest are the variation in the initial area difference and the variation in the Page time. Respectively we recall them to be,

$$\delta(\Delta\mathcal{A}(0)) = \Delta\mathcal{A}(0)|_{\phi_0/T\approx35.0} - \Delta\mathcal{A}(0)|_{\phi_0/T=0}, \tag{77}$$

$$\delta t_p = t_p|_{\phi_0/T\approx35.0} - t_p|_{\phi_0/T=0}, \tag{78}$$

and these are plotted as functions of $x_{\mathcal{R}}$ (in units of $r_+$) in Figure 11.

We again look at (75) in $r_+ = 1$ units. To reiterate, the variation in the initial area surface can be thought of as coming from the backreaction in the exterior, whereas $\delta v$—the variation in entanglement velocity—comes from the backreaction of the interior. For $x_{\mathcal{R}} \approx x_p$, the first

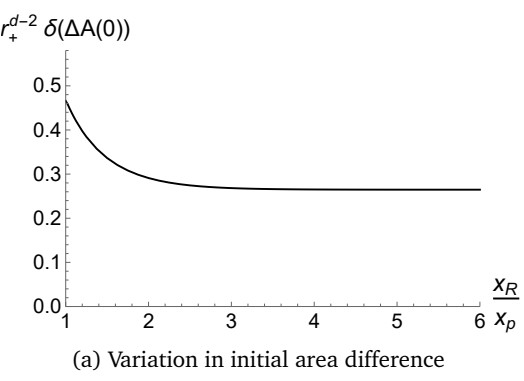
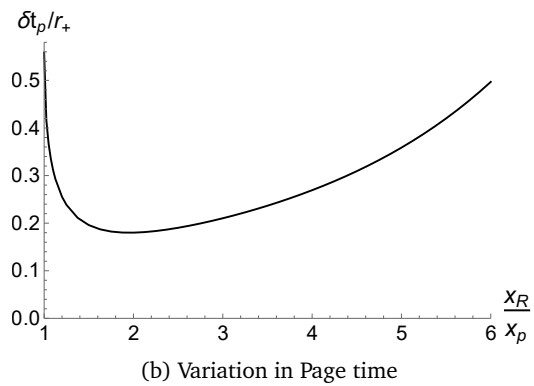

(a) Variation in initial area difference

(b) Variation in Page time

Figure 11: The variations in the (a) initial area difference and the (b) Page time as functions of the radiation region. We compute these quantities up to $x_{\mathcal{R}}/x_p = 6$, at which point our numerics become too noisy to be taken as reliable. Nonetheless, even within this range the numerics indicate that the variation in initial area difference monotonically falls while the variation in the Page time drops then rises.

---

[18]We reiterate that there technically exists a Neumann $\Delta = d/2$ quantization which we do not include here.

term is much closer to the order of $\delta t_p$ than the second term. This is seen in our previous analysis and again in the early part of the plots in Figure 11.

However, taking the behavior in Figure 11 seriously would indicate that $\delta(\Delta\mathcal{A}(0))$ remains $O(10^{-1})$ even as $x_{\mathcal{R}} \gg x_p$, whereas $\delta t_p$ increases and is eventually orders of magnitude larger. (75) would imply that the $\delta\nu$ term is responsible—specifically the initial area difference $\Delta\mathcal{A}(0)$ itself is what grows and thus more heavily weights the effect of $\delta\nu$ on $\delta t_p$.

Physically, we interpret this is as the existence of two regimes: $x_{\mathcal{R}} \approx x_p$ in which the exterior backreaction plays a significant, leading-order role in determining the Page time and $x_{\mathcal{R}} \gg x_p$ in which the interior backreaction has a stronger influence on Page time. Nonetheless these numerics still indicate that, regardless of the regime of $x_{\mathcal{R}}$, the Page time with a scalar deformation is longer than with no scalar deformation.

## 4  Conclusions and Outlook

In this paper, we use a doubly holographic setup to investigate the effect of a bath deformation on the fine-grained entanglement dynamics of a gravitating black hole. We add a relevant scalar operator in the bath and study the change of the Page curve, Page point, and Page time as functions of the deformation parameter $\phi_0/T^{d-\Delta}$. Performing two analyses, we find the following:

(1) Keeping the radiation region $\mathcal{R}$ fixed, we increase the deformation parameter and find, within our numerics, that the Page time monotonically increases. Similarly, the initial area difference also increases, so scalar backreaction on the geometry both outside and inside of the black hole informs the Page curve.

(2) Comparing two fixed values of the deformation parameter, we find that the difference in Page time increases as $\mathcal{R}$ is pushed away from the interface with the brane, i.e. as we trace out more boundary degrees of freedom. The change in the initial area difference meanwhile levels-off, so backreaction in the interior—the trans-IR effects—appears to become more important.

There are several different aspects to the physics of our setup and findings. We list them below, as well as some potential avenues of future exploration.

### Parameterization of Coarse-Graining

We observe that for any generic value of the deformation parameter $\phi_0/T^{d-\Delta}$, the fine-grained entropy curve for the black hole is accessible, even though the bath undergoes some coarse-graining. Increasing $\phi_0/T^{d-\Delta}$ affects the radial scale, denoted by $r_{\mathrm{RG}}$, at which the scalar backreaction becomes significant.

This radial scale can be identified with an energy scale in the bath; we denote this by $E_{\mathrm{RG}}$. For any fine-grained physical question above this energy scale, the UV CFT state should be a good description, while for more coarse-grained physics below this scale, IR/trans-IR states reached by the holographic RG flow should dominate.

That being said, given the RT surface ending on the brane, its intersection with the brane $r_T$ can be compared to $r_{\mathrm{RG}}$. These are two independent scales which define separate procedures. While $r_{\mathrm{RG}}$ defines the regime of dominance of the UV fixed point in the holographic RG flow, $r_T$ is related to the size of the radiation region by (49) and is thus reflective of how much of the bath is traced out.

Comparing these two scales can shed light on whether there is any sense in which the island could be made ignorant about the scalar backreaction, in terms of (1) when the entanglement

wedge contains a part of the black hole interior and (2) whether the island depends crucially on the coarse-graining of the bath. On the other hand, by construction, the Hartman-Maldacena surface can never be ignorant about the scalar backreaction, since it probes the entire geometry. There is thus a question of which part of the Page curve is affected more by the tracing-out of the bath versus the coarse-graining tied to the RG flow, particularly as the deformation is tuned. We have most directly addressed this question, but a simple and potentially informative extension to our analysis would be to keep $r_T$ fixed (which would lead to $x_{\mathcal{R}}$ changing with the deformation) and study the resulting behavior of the Page time.

### RG-Flowing the BCFT State

If we put aside the black hole information paradox, our results can be interpreted purely in terms of the boundary degrees of freedom and are relevant for our understanding of the QFT dynamics at strong coupling. Note that here we are momentarily adopting the vocabulary of [53]; "bulk" in the BCFT context refers to the continuous $d$-dimensional CFT degrees of freedom while "boundary" refers to the degrees of freedom localized to the $(d-1)$-dimensional "defect" CFT. We start with the expectations set by a simple model which, although potentially different from our setup, is nonetheless instructive.

Usually in the BCFT context, an RG flow is considered within the CFT bulk degrees of freedom, keeping the boundary unaltered.[19] However, the bulk RG flow gets related to an RG flow in the boundary conditions such that the corresponding boundary states change. In fact, mapping out all possible fixed points under all possible relevant deformations to a CFT is equivalent to classifying all possible boundary conditions for that CFT; see [64] for a detailed proposal regarding this point.

If we imagine turning on the relevant deformation to the bulk CFT (the bath) at the same time as coupling it with the boundary CFT, a coarse-grained count of the microstates in the boundary CFT can be obtained by computing entanglement entropy and waiting sufficiently long in time. The Page time is then governed by the ratio of central charges $c_{\text{bdry}}/c_{\text{bulk}}$ as demonstrated by [53].

Along an individual real RG flow of the bulk CFT, $c_{\text{bulk}}$ decreases by the usual $c$-theorem. If the boundary CFT is indeed kept unaltered—so $c_{\text{bdry}}$ stays fixed—then the ratio of central charges increases under the RG flow. The result would then be an increasing Page time as the deformation parameter is increased, assuming the UV physics is dominant or the IR is stable with respect to $\phi_0/T^{d-\Delta}$.[20] This is because larger values of $\phi_0/T^{d-\Delta}$ correspond to RG flows for which $r_{\text{RG}}$ approaches the boundary, thus the IR physics—which sets a minimum $c_{\text{bulk}}$—dominates.

However, note that our procedure does not actually keep the boundary CFT fixed. There is a nontrivial tower of scalar fields localized to the brane which should then be dual to a nontrivial tower of scalar operators in the boundary CFT—such operators arise naturally from the bulk CFT operator by a *boundary operator product expansion (BOPE)* [65]. If the boundary-localized modes are irrelevant or even marginally irrelevant, then this is not a problem; the boundary CFT is still not undergoing a nontrivial RG flow and being coarse-grained. Nonetheless, it is possible that these modes are relevant in our setup. Regardless, within the regime probed by out numerics, we still see an increase in the Page time, indicating two possibilities:

---

[19]Note, however, that we are considering the RG flow of a state in the theory and not the theory itself. While we do not associate a central charge with the particular state of the system, it is nevertheless true that the IR description is arrived at by integrating out UV modes. Therefore, the accessible degree of freedom, even for the given state, is expected to decrease.

[20]The subtlety here lies in the UV fixed point being the same for all of the flows labeled by the deformation parameter, but the IR points generically being different. If the UV physics dominates, then these differences do not matter. If the UV physics does not dominate, then a lack sensitivity of the IR regimes to $\phi_0/T^{d-\Delta}$ would allow for increasing Page time, since $r_{\text{RG}}$ is still monotonically approaching the boundary.

(1) For some reason, $c_\text{bulk}$ decreases faster than $c_\text{bdry}$, such that $c_\text{bdry}/c_\text{bulk}$ ultimately increases anyway.

(2) While our discussion assumes that the ratios of central charges provide good, leading-order approximations for the Page time, perhaps the real story is more complicated and the Page time depends more nontrivially on other parameters.

Additionally, we reiterate that there is an analytically-continued part of the RG flow beyond the IR, as well. This trans-IR flow is also probed by the Page curves (albeit only up to $r_*$ (61)) and could further complicate the intuition gained from the simpler model.

Our observation regarding the lack of sensitivity of the Page time to $\Delta$ for large deformation parameter is consistent with the Page time being determined by a ratio of the central charges, and for a massive QFT as a bath, in the infinite deformation limit, depends only the ratio of the number of degrees of freedom.

### Large Deformations and Significance of the Page Point

We also note the physical significance of the Page point; it essentially sets a critical lower bound for the degrees of freedom which must be traced out to see nontrivial time-dependence in the Page curve. Our numerical observations suggest that as $\phi_0/T^{d-\Delta}$ is tuned to larger values, the Page point seems to decrease and ultimately asymptote to a nonzero value in the supergravity limit.

*A priori* there are three possibilities for the asymptotic behavior[21] of the Page point as $\phi_0/T^{d-\Delta} \to \infty$. It may either (1) approach 0, (2) indeed level-off at a finite value, or (3) increase without bound. Each result would have its own physical interpretation and potential questions:

(1) For large deformations, the need to trace out the bath would become less and less significant. We would observe the full fine-grained Page curve of the Hawking radiation while not needing to integrate out bath degrees of freedom after taking $\phi_0/T^{d-\Delta} \to \infty$.

(2) There would be a nontrivial lower bound for the amount of tracing-out needed to observe a Page curve. It would then be natural to wonder what parameters of the theory set such a value.

(3) For large deformations, we would essentially need to trace out all bath degrees of freedom to see a Page curve. However in this extreme limit, we do not actually expect to see a Page curve because, without a bath, Hawking radiation simply leaves the gravitating system due to the transparent boundary conditions at the interface. We thus get a nonunitary entropy curve and therefore an information paradox in the $\phi_0/T^{d-\Delta} \to \infty$ limit. There would also necessarily be a minimum Page point for some finite deformation—the tuning of this value would be important to understand.

It would thus be interesting to perform an agnostic analysis to figure out which of these three pictures actually happens. Relatedly, our numerics also suggest that the Page time becomes independent of $\Delta$, as long as $\Delta < d$, in the $\phi_0/T^{d-\Delta} \to \infty$ limit. However, we require more robust numerics capable of probing larger $\phi_0/T^{d-\Delta}$ to state this conclusively.

---

[21]There could also be interesting behavior leading up to the asymptotic limit, such as saddles, but what such behavior would mean in principle is unclear.

**Future Directions**

There are several additional questions related to our work beyond those mentioned above that are worth exploring:

- In our setup, the scalar field $\phi$ dual to the boundary operator is supported both on the bulk and on the brane. Thus, our results encode a deformation of the bath and the brane. To isolate the effects on the bath we need a scalar field $\tilde{\phi}(r, x^1)$ such that $\tilde{\phi}$ vanishes at $x^1 = 0$. Then in the brane plus bath theory, we will only be deforming the bath. The brane physics could still be influenced through the scalar field in the extra dimension, but a braneworld-restricted observer would be expected to be blind to it.

- Adding higher-curvature terms, either in induced gravity or as done with JT/DGP gravity [14], would change the theory on the brane and would allow $d = 2$ to be studied. Another possibility is to add higher-order interaction terms to the scalar action. Studying such Kasner flows is interesting in itself, beyond the context of entanglement islands, since there are hints of universal behavior when one considers more general scalar actions [55].

- In this work, we have considered tensionless branes. Extending our work to nonzero-tension branes adds an extra parameter to the picture. Such work, however, would likely require a modified ansatz and more intricate numerics, but this would be a necessary step in extending our exploration to the case of near-critical braneworlds where the localization of gravity is more "sharp" [11].

  Due to existence of critical brane angles [15, 16, 27, 66] dictating the appearance of a Page curve as $x_{\mathcal{R}} \to 0$, we expect that any sufficiently large (but still subcritical) tension would eliminate the Page point entirely. Thus, we should be able to take $x_{\mathcal{R}} \to 0$ for braneworlds with "sharper" localization of gravity.

- We consider higher-dimensional braneworld constructions, but a natural question would be to examine the other well-studied class of toy models—2-dimensional dilatonic gravity. A natural follow-up would be to deform the bath in the original 2-dimensional evaporation model [3] so as to study the dependence of the Page time on the deformation. Another follow-up in this vein but closer to our own work would be to deform the bath in the 2-dimensional eternal model [19].

- It would be interesting to sharpen the observation of how the Page time depends on the ratio of the bulk and the boundary CFT central charges, specifically along an RG flow in the bulk CFT. This calculation can be performed from a top-down model by considering Karch-Randall branes in a supergravity geometry (as in [66]) that encodes such an RG flow.

- That our results are done in a supergravity limit makes the following question natural: What happens if we perform an RG flow in the bath, going away from the supergravity limit? It would be particularly interesting to study the asymptotics of both the Page point and the Page time.

- We may ask how rotation affects the picture. Note that solving for the KR brane in even a vacuum solution with nontrivial angular momentum and working through the standard island construction would be an interesting avenue to pursue. Rotation would allow for a broader class of flows.

- We can add to the bulk a $U(1)$ gauge field coupled to the scalar field, such that the gauge symmetry spontaneously breaks. This effect is *holographic superconductivity*. It

is interesting to ask how it may be probed by Page times. Note that Kasner flows have been linked to holographic superconductivity by [67].

We hope to return to some of these questions in the near future.

## Acknowledgements

We thank Sean Hartnoll, Andreas Karch, and Jorrit Kruthoff for reading the draft. We also thank Andreas Karch and Chethan Krishnan for additional comments and very useful discussion. EC and SS are supported by National Science Foundation (NSF) Grant No. PHY-1820712. SS is also supported by NSF Grant No. PHY–1914679. AK acknowledges support from the Department of Atomic Energy, Govt. of India and IFCPAR/CEFIPRA, project no. 6403. AKP is supported by CSIR Fellowship No. 09/489(0108)/2017-EMR-I.

## A    Ambiguity in the Dual Operator Dimension

We can generically consider the near-boundary ($r \to 0$) mode expansion of a scalar field with in $AdS_{d+1}$ to be,

$$\phi(r) \sim \phi_- r^{\Delta_-} + \phi_+ r^{\Delta_+} , \tag{79}$$

where $\Delta_\pm$ are distinct roots of (10),

$$\Delta_\pm = \frac{1}{2}\left(d \pm \sqrt{d^2 + 4m^2}\right) . \tag{80}$$

From the mass-dimension relation, the conformal dimension $\Delta$ of the scalar operator will be relevant (assuming it is physical—$\Delta > 0$), i.e.,

$$\Delta < d , \tag{81}$$

if and only if $m^2 < 0$. However, we must also consider the Breitenlohner-Freedman stability bound [57],

$$-\frac{d^2}{4} \leq m^2 < 0 . \tag{82}$$

Meanwhile the roots (80) satisfy,

$$0 < \Delta_- \leq \frac{d}{2} \leq \Delta_+ < d , \tag{83}$$

with $\Delta_\pm = d/2$ when $m^2 = -d^2/4$. Thus when this bound is satisfied, (79) breaks down. Above the bound, modes in (79) are normalizable.

Based on value of $m^2$, there is a choice to be made for which root is taken to be $\Delta$ (related to the boundary conditions of $\phi$ [59]). The only choice for $m^2 \geq 1 - d^2/4$ is $\Delta = \Delta_+$ because selecting $\Delta = \Delta_-$ would violate the unitarity bound,

$$\Delta > \frac{d-2}{2} . \tag{84}$$

Meanwhile, $m^2 < 1 - d^2/4$ gives us both as options [58]. While the canonical choice $\Delta = \Delta_+$ used by [24] restricts us to $\Delta \geq d/2$ (stricter than unitarity), using the quantization for which $\Delta = \Delta_-$ instead allows us to reach $(d-2)/2$. So for a bulk scalar field with

$-d^2/4 < m^2 < 1-d^2/4$, we have two CFTs—one with $\Delta = \Delta_+$ and one with $\Delta = \Delta_-$—related by a Legendre transform of the generating functional.

For $\Delta = \Delta_+$, in (79) we now identify the leading-order coefficient $\phi_-$ as the source $\phi_0$ and the next-to-leading-order coefficient $\phi_+$ as (proportional to) the one-point function $\langle \mathcal{O} \rangle$,

$$\phi(r) \sim \phi_0 r^{\Delta_-} + \frac{\langle \mathcal{O} \rangle}{2\Delta_+ - d} r^{\Delta_+} . \tag{85}$$

However this interpretation is flipped if $\Delta = \Delta_-$. Now the $\phi_+$ is the source while $\phi_-$ is proportional to the one-point function. Either way, a generic way to write the near-boundary expansion is as,

$$\phi(r) \sim \phi_0 r^{d-\Delta} + \frac{\langle \mathcal{O} \rangle}{2\Delta - d} r^{\Delta} , \tag{86}$$

since $\Delta_+ + \Delta_- = d$. We use this expression in the main text.

As for $\Delta = d/2$, there is a divergence which appears in the $r^\Delta$ term, making the breakdown of this mode expansion more evident. [59] resolves this problem by performing an expansion of the relevant Bessel function $K_0$, ultimately writing,

$$\phi(r) \sim \phi_0 r^{d/2} \log r . \tag{87}$$

## B  Numerical Determination of Emergent $p_t$

The numerical plots are obtained by shooting the radial functions from the horizon $r = r_+$ to both the boundary $r \to 0$ and the singularity $r \to \infty$.[22] By assuming regularity at the horizon, we may expand $\phi$, $f$, and $\chi$ as,

$$\phi(r) = \phi_+ + \phi'_+(r - r_+) + O[(r - r_+)^2], \tag{88}$$

$$f(r) = f'_+(r - r_+) + O[(r - r_+)^2], \tag{89}$$

$$\chi(r) = \chi_+ + \chi'_+(r - r_+) + O[(r - r_+)^2]. \tag{90}$$

The subscript $+$ denotes values at the horizon as in (23). Plugging these into the equations of motion (11)-(13)[23] and taking $r \to r_+$, we have the constraints,

$$0 = \frac{\Delta(d - \Delta)\phi_+}{r_+} + r_+ f'_+ \phi'_+ , \tag{91}$$

$$0 = -\frac{\Delta(d - \Delta)\phi_+^2}{d - 1} - 2(d + r_+ f'_+), \tag{92}$$

$$0 = \frac{r_+(\phi'_+)^2}{d - 1} - \chi'_+ . \tag{93}$$

We can solve these equations to obtain the series coefficients,[24]

$$\phi_+ = \mp \frac{i\sqrt{2}\sqrt{d-1}\sqrt{d + f'_+ r_+}}{\sqrt{\Delta(d - \Delta)}} , \tag{94}$$

$$\phi'_+ = \pm \frac{i\sqrt{2}\sqrt{d-1}\sqrt{d + f'_+ r_+}\sqrt{\Delta(d - \Delta)}}{f'_+ r_+^2} , \tag{95}$$

$$\chi'_+ = -\frac{2(d + f'_+ r_+)[\Delta(d - \Delta)]}{f'^2_+ r_+^3} . \tag{96}$$

---

[22]Technically, we integrate up to cutoffs near these limits to obtain our plots.

[23]We multiply (11) and (12) by $rf(r)$ so as to avoid poles and obtain a finite result.

[24]There is a branching of the $\phi_+$ and $\phi'_+$ coefficients. Note that these branches go together. For example the $-$ expression for $\phi_+$ is paired with the $+$ expression for $\phi'_+$.

Even with these coefficients, we still have the freedom to set a scale by numerically fixing $f'_+$ so long as we keep it negative. In doing so, we further set $\chi_+ = 0$. Then for each value of $r_+$ and taking some comparatively small $\epsilon > 0$, we can integrate the radial functions either from $r = r_+ - \epsilon$ (outside of the horizon) to the boundary or from $r = r_+ + \epsilon$ (inside of the horizon) to the singularity.

By integrating to the boundary, we obtain $\phi(r)$ and $\chi(r)$ in the exterior. The field is used to get $\phi_0$, but how we do so depends on whether $\Delta > d/2$ (the $\Delta = \Delta_+$ quantization) or $\Delta < d/2$ (the $\Delta = \Delta_-$ quantization). This is because the power of the source term is only leading ($d - \Delta < \Delta$) in the former case. From (18) we find,

$$
\phi_0 =
\begin{cases}
\displaystyle\lim_{r \to 0} r^{\Delta-d} \phi(r), & \text{if } \Delta > \dfrac{d}{2}, \\[2ex]
\displaystyle\lim_{r \to 0} -\frac{r^{2\Delta-d+1}}{2\Delta-d} \partial_r \left[ r^{-\Delta} \phi(r) \right], & \text{if } \Delta < \dfrac{d}{2}.
\end{cases}
\tag{97}
$$

The different branches of (94) and (95) will yield either $\phi_0 > 0$ or $\phi_0 < 0$. We are concerned with the former case and neglect the latter. The relevant branch to obtain a positive source depends on whether $\Delta > d/2$ or $\Delta < d/2$.

We remark that this method breaks down for $\Delta = d/2$ because the expansion (18) also breaks down. We instead use the (Dirichlet) logarithmic expression [59],

$$
\phi_0 = \lim_{r \to 0} \frac{r^{-d/2}}{\log r} \phi(r), \quad \text{if } \Delta = \frac{d}{2}.
\tag{98}
$$

For $\chi(r)$ in the exterior, as we set $\chi_+ = 0$ at the horizon, $\chi(0)$ may not be 0 despite this being the expected near-boundary behavior—we have solved for $\chi(r)$ backwards. However by simply evaluating $\chi(0)$ and shifting the entire function by this amount, we can obtain the "true" $\chi(r)$ for which $\chi(0) = 0$. In doing so, we also obtain the "true" $\chi_+$ and thus the temperature $T$.

When integrating to the singularity, we obtain $\phi(r)$, from which we extract the coefficient $c$ in (24). We then use $c$ to obtain the Kasner exponent $p_t$.

For each $r_+$, we get a particular ordered pair $(\phi_0/T^{d-\Delta}, p_t)$ of dimensionless quantities. By plotting the interpolating functions for a large number of points, we obtain Figure 3. Additionally by numerically computing the radial functions in this manner, we can compute geometric quantities such as the area of RT surfaces.

## C  Initial Hartman-Maldacena Area in AdS-Schwarzschild

It is a straightforward exercise to compute the area of the Hartman-Maldacena surface at $t = 0$ analytically when the geometry is AdS-Schwarzschild. We present the calculation here.

For a $(d + 1)$-dimensional AdS-Schwarzschild black hole with blackening factor $f(r) = 1 - (r/r_+)^d$, the integral (50) becomes,

$$
\mathcal{A}_{HM}(0) = 2 \int_0^{r_+} \frac{dr}{r^{d-1} \sqrt{1 - (r/r_+)^d}} = \frac{2}{r_+^{d-2}} \int_0^1 \frac{d\tilde{r}}{\tilde{r}^{d-1} \sqrt{1 - \tilde{r}^d}}, \quad \tilde{r} = \frac{r}{r_+}.
\tag{99}
$$

The antiderivatives of the integrand for $d > 2$ and $d = 2$ are,

$$
2 \int \frac{d\tilde{r}}{\tilde{r}^{d-1} \sqrt{1 - \tilde{r}^d}} =
\begin{cases}
-\dfrac{2}{d-2} \dfrac{1}{\tilde{r}^{d-2}} \, {}_2F_1\left(\dfrac{1}{2}, \dfrac{2-d}{d}; \dfrac{2}{d}; \tilde{r}^d\right) & \text{if } d > 2, \\[2ex]
-2 \text{Tanh}^{-1} \sqrt{1 - \tilde{r}^2} & \text{if } d = 2.
\end{cases}
\tag{100}
$$

However both diverge for $\tilde{r} = 0$. Taking a cutoff $\tilde{r} = \epsilon \ll 1$, the respective divergent terms in (99) take the form,

$$\frac{2}{d-2}\frac{1}{\epsilon^{d-2}}\,, \quad \text{if } d > 2\,, \tag{101}$$

$$-2\log\epsilon\,, \quad \text{if } d = 2\,, \tag{102}$$

which are both precisely canceled by the counterterms (52). Thus after renormalizing, the initial Hartman-Maldacena surface areas are,

$$r_+^{d-2}\mathcal{A}_{HM}(0) = \begin{cases} -\dfrac{2\sqrt{\pi}\,\Gamma\!\left(\frac{2}{d}\right)}{(d-2)\Gamma\!\left(\frac{4-d}{2d}\right)}\,, & \text{if } d > 2\,, \\[3mm] \log 4\,, & \text{if } d = 2\,. \end{cases} \tag{103}$$

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
