# Peer review of "Page Curves and Bath Deformations"

_SciPost Physics, doi:SciPost Phys. Core 5, 033 (2022)_

## Round 1 · Referee Report · Anonymous (Referee 1) · 2022-1-31

Report

This paper investigates the behavior of the Page curve in doubly-holographic models of black hole evaporation, focusing on the effect of turning on a relevant scalar deformation in the bath into which the Hawking radiation is escaping. The (doubly) holographic dual to turning on such a deformation is a holographic RG flow geometry. In order to investigate the behavior of the Page curve in a model of semiclassical black hole evaporation in this context, the paper considers introducing a tensionless end-of-the-world brane in the holographic RG flow geometry that connects the two asymptotic boundaries of the asymptotically AdS geometry. Because this brane slices through the black hole portion of the space time, the paper claims that the brane describes an effective gravitational theory containing an evaportating black hole. Computing the entropy of the Hawking radiation of this black hole amounts to computing the entropy of regions of the boundary of the holographic RG flow geometry, which is computed by HRT surfaces that may or may not end on the brane. The transition in the Page curve corresponds to an exchange of dominance between HRT surfaces that do not end on the brane and those that do, and in this context the paper examines how changing the Page time depends on quantities like the strength of the scalar deformation or the position size of the radiation region whose entropy is being computed.

The investigation of the structure and origin of the Page curve in black hole evaporation is extremely topical and of great interest to much of the quantum gravitational community. Moreover, in certain contexts doubly-holographic models may indeed be interesting test grounds in which to perform such analyses. However, I must confess that it is not clear to me how the present paper substantially answers any prescient questions in the context of black hole evaporation, and while I think all the computations in the paper are correct, the particular model of double holography being used here is sufficiently far removed from typical semiclassical models of gravity that it's difficult to imagine what lessons drawn from this model, if any, apply to theories of gravity of more direct interest (like general relativity). To specifically address SciPost's acceptance criteria: I do not think that this paper detail a groundbreaking theoretical/experimental/computational discovery, presents a breakthrough on a previously-identified and long-standing research stumbling block, or provides a novel and synergetic link between different research areas. It could potentially open a new pathway in an existing or a new research direction, with clear potential for multipronged follow-up work, but if that is the goal of the paper then substantial modifications are needed to address clearly what new pathway is being opened. For these reasons, I cannot recommend publication of the paper in SciPost without at the very least a major revision.

Requested changes

To be more explicit, the points of most concern are the following: 1. In the recent models of BH evaporation, the external bath coupled to the gravitational system is just a proxy for the Hawking radiation: it allows us to unambiguously separate the radiation from the BH. However, ideally we would like to understand the process of BH evaporation entirely in the gravitational context, with no need for an auxiliary reservoir at all. So how does modifying properties of the reservoir (such as turning on a relevant deformation) teach us more about the fundamental physics of the evaporation process itself, rather than merely about the way in which we've simplified the evaporation process for our convenience? Perhaps more operationally, what would "deforming the bath system" mean in the context of the evaporation of an asymptotically flat BH?

  1. Because the KR brane used in this paper is tensionless, the computation of HRT surfaces in this paper essentially amounts to computations of the entanglement entropy of two strips (centered around the plane $x^1 = 0$) in the thermofield double state of two copies of a holographic CFT with some relevant scalar deformation turned on. (That is, the KR brane is located on the surface $x^1 = 0$, which is a surface of $\mathbb{Z}_2$ symmetry, so computations of HRT surfaces in the presence of the brane, with Neumann boundary conditions at the brane, are equivalent to computations of HRT surfaces in the full geometry with no brane.) So it seems to me that the "Page curve" being computed here is simply the Hartman-Maldacena transition, except in an RG flow geometry.

  2. Related to the above point, it's true that once can quotient by the $\mathbb{Z}_2$ symmetry to create an end-of-the-world brane at $x^1 = 0$ and then try to interpret this brane as supporting an evaporating black hole. However, I'm not convinced that the "effective gravitational theory" induced on the tensionless KR brane will necessarily tell us much about the behavior of more conventional theories of semiclassical gravity. I think the paper does a very nice job at the beginning of Section 3 discussing the way in which gravity localizes to the KR brane at the critical tension, and is quite explicit about how the tensionless limit most removes the notion of gravity "localizing" to the brane. I appreciate this clarification, but I wonder what we're really learning, then, from the tensionless limit. Why does the paper not consider branes at the critical tension, perhaps then searching for bulk HRT surface numerically? Why not consider turning on a deformaion in the bath CFTs of the original BH evaporation model of Almheiri, Engelhardt, Marolf, and Maxfield to see explicitly what happens in a semiclassical model, without the need for a doubly-holographic setup?

  3. Finally, the behavior of the Page curve as the size of the radiation system is modified makes me question how good the separation between the "black hole" and the "radiation" system actually is in this model. Typically, one is interested not just in the entropy of //some// portion of the radiation, but of //all// the radiation. But in the model considered in this paper, the entropy of all the radiation corresponds to moving the point $x_\mathcal{R}$ all the way to $x^1 = 0$, and this always gives no Page curve, regardless of the strength of the deformation. (Contrast this with, say, the East Coast evaporation model of Almheiri, Hartman, Maldacena, Shaghoulian, and Tajdini, where taking the limit of $x_\mathcal{R}$ approaching the boundary of the radiation reservoir still gives a Page curve.) The fact that one must necessarily consider only a "subsystem" of the reservoir in order to obtain a Page curve suggests to me that the "black hole" system really bleeds into the reservoir a little bit -- conceivably this could be related to the lack of localization of the gravitational theory on the tensionless brane. So without a clear-cut distinction between the radiation and the black hole systems, is the Page curve being examined in this paper really the Page curve of //the// radiation, or merely the Page curve of some partitioning of some quantum system that contains something like a black hole somewhere and something like radiation somewhere else?

  • validity: high
  • significance: ok
  • originality: low
  • clarity: good
  • formatting: perfect
  • grammar: perfect

Author:  Sanjit Shashi  on 2022-04-19  [id 2395]

(in reply to Report 1 on 2022-01-31)

We thank the referee for their response. Before we dig into the specific points raised, we would like to mention some broader points regarding our work and its presentation:

  • We have introduced subsections to the Introduction, so as to help with the pacing and better highlight the subtleties of our discussion.

  • The statement that the model we consider is far removed from something like GR is a criticism that can be applied to essentially all of the models studied in the context of islands. Indeed, the gravitational models in which the island rule has been most firmly proven in a variety of ways are in 2-dimensional gravity, with the tools (replica wormholes [arXiv:1911.12333] and thermodynamical approaches using the nonlocal Polyakov action [arXiv:2107.10358, arXiv:2111.06912]) being distinctly 2d features. Even the higher-dimensional braneworld models mostly feature massive gravity [arXiv:2006.02438], and although it is possible to construct massless gravity by employing a gravitating bath [arXiv:2012.04671], the result is that there is no phase transition (although, we acknowledge that whether the entropy computed therein is the "correct" way to measure Hawking entropy is a matter of debate [arXiv:2103.17253]). Ultimately, both 2d models and higher-dimensional models employing branes and double holography are just toy models which each have features that make them distinct from "typical" gravity. We believe this to be a key point, so we have added some of this discussion to our Introduction, including a new paragraph on page 2 starting with, "The island rule has been studied so far..."

  • Just to be clear, the objects of interest in this paper are not evaporating black holes, but eternal ones which are in thermal equilibrium with their baths. A version of the information paradox in such systems was explored by [arXiv:1910.11077], and this has formed the basis for the work in higher-dimensional braneworlds [arXiv:1911.09666, arXiv:2006.04851, arXiv:2010.00018, arXiv:2006.02438, arXiv:2012.04671]. We have further emphasized this subtlety throughout the Introduction by putting it all into a new Section 1.2 starting on page 4.

Responding to each of the numbered points:

(1) We agree that the nongravitating bath is ultimately a computational tool, and not a desirable feature. The point of the discussion in the Introduction (particularly in the new Section 1.3) is to present the previous arguments that introducing the bath is "too much" of an aid, and also to motivate deviations away from the basic setups using it. However, such statements are simply about our underlying motivations---we make no claims about understanding the evaporation process without a nongravitating bath in this paper. Instead, we opt for an alternate analysis---to slightly affect the bath in order to understand how even that might play a role in the higher-dimensional island story.

For clarity, we have added text (particularly the paragraph starting, "As a first step..." on page 2 and the first sentence of Section 1.4 on page 6, "In this work...") that states that we are merely affecting the bath, not eliminating it. In the end, our results show that the bath is not a consistent computational tool.

(2) We agree with the referee that this is another way to interpret the Page curves in our work, but the whole point of double holography is that classical entanglement surfaces in the bulk are, to leading order in $1/G_N$, quantum extremal surfaces in the braneworld theory. Thus, the QES phase transition in the braneworld can be seen to leading order in perturbation theory as the classical phase transition (the HM transition) in the higher-dimensional bulk, with the latter being much easier to compute.

Of course as noted by the referee, in an orbifolded geometry the latter transition is quite literally just the Hartman-Maldacena phase transition quotiented by parity. We now explicitly note this---see on page 5 the two paragraphs which begin with "Note that this phase transition..." However, this is the purely classical bulk interpretation, whereas the key picture is that of the braneworld theory [arXiv:2006.02438].

(3) Regarding the first question, we work with the tensionless case for simplicity. In particular, the boundary condition of the scalar field is much simpler---it is trivially satisfied by our ansatz if the brane is tensionless. Furthermore, the point of the discussion on KR branes is not that "localization" is removed. Rather, it is that there are different levels of sharpness to the localization of gravity, and that there is still a perfectly valid interpretation of gravity on the brane even in the tensionless regime (albeit one which is highly non-standard). In short, we follow [arXiv:2006.02438] in arguing that tensionless braneworlds support some theory of induced gravity. We discuss this both in page 3 ("A particularly useful manifestation...")

Ultimately, we are only using the tensionless construction as a preliminary toy model. We do believe that an important next step is to consider the near-critical regime. We have elaborated upon this, including some of the expected complications, in the Future Directions (page 32, "In this work, we have considered tensionless branes...").

As for deforming the bath in the original AEMM model [arXiv:1905.08762], such an approach would fall outside of the realm of double holography, which is why we do not consider it. However, this is also an important follow-up which would synthesize our core idea with the other class of well-studied toy models (2d dilatonic gravity), and so we have added it to the Future Directions (page 33, "We consider higher-dimensional braneworld constructions..."). A follow-up closer to our work (i.e. considering the eternal case instead of the evaporating case) would be to deform the bath in the eternal model of [arXiv:1910.11077].

(4) First, let us note that the gravitational system \textit{certainly} bleeds into the bath a bit. This is why there is a massive graviton in the first place---the stress tensor is not conserved at the interface between the bath and the brane due to the transparent boundary conditions. Such a statement is true even with tensionful branes. As an example (which we admit is in a different background but is still concerned with the same base question of "For what $x_{\mathcal{R}}$ do we have a Page curve?"), see [arXiv:2112.09132], Figure 7, in which one sees that taking $x_\mathcal{R}$ (there represented as $\Gamma$) sufficiently close to the brane-bath interface for small positive tension leads to a loss of the Page curve.

Now, whether this is to be interpreted as the Page curve of Hawking radiation is a matter of interpretation. In following the past literature on double holography [arXiv:1911.09666, arXiv:2006.04851, arXiv:2010.00018, arXiv:2006.02438], we would say that it is, but one is certainly allowed to take issue with this interpretation. This is a matter of philosophy, however, and a different philosophy would require a different entropy proposal altogether. We add some discussion to our Introduction to that effect (page 4, "We note that interpreting...").

We do believe that this question inspires additional statements on one of the Future Directions suggested by the referees. One of the features of the doubly holographic models seen not just in [arXiv:2112.09132], but also in [arXiv:2012.04671, arXiv:2010.00018], is that there is a finite tension beyond which the Page point indeed goes to the interface. In particular, this would mean that for near-critical tensions, we can indeed take $x_\mathcal{R} \to 0$ and expect a Page curve, so we wonder whether there is some notion of localization coinciding with the ability to take $x_\mathcal{R} \to 0$. Simply put, while the existence of a Page point is not simply a result of us working with a tensionless brane (since it should appear for a range of finite-tension branes, as well), we believe that it does not appear for the near-critical regime. We have added this statement to our Future Directions (page 33, "Due to existence of critical brane angles...").

---

## Round 1 · Referee Report · Michal P. Heller (Referee 2) · 2022-4-9

Report

I read the paper and the report of another referee. I second another referee's points -- they are important and should be clarified. I do not have points that I would like to add myself.

Now, regarding the acceptance criteria, I agree with the other referee that the paper, as it is, does not introduce a breakthrough advancement to the field. However, testing the robustness of existing predictions with respect to relaxing underlying symmetries constitutes a priori an important research direction. It is perceived as incremental when no dramatic alterations of the existing messages occur, an interesting advancement when a qualitatively new effects arise and a breakthrough if fine tuning turns out to be crucially needed for a physics effect in question. To my understanding, the present paper lies somewhere between an incremental advancement and studying new interesting effects. So given very strict SciPost acceptance criteria my suggestion to the authors would be to try to better motivate why the effects associated with relevant bath deformations are interesting and important from the point of view of developing the field of islands and Page curves.
  • validity: high
  • significance: ok
  • originality: ok
  • clarity: high
  • formatting: perfect
  • grammar: perfect

Author:  Sanjit Shashi  on 2022-04-19  [id 2396]

(in reply to Report 2 by Michal P. Heller on 2022-04-09)

We thank Dr. Heller for his response. As discussed in our response to referee #1, we expanded upon the motivation in the beginning of our Introduction (page 2) in the hope of better explaining how our work adds to the story of entanglement islands, as well as clarifying how adding a relevant deformation to the bath is a useful deviation from previously explored doubly holographic setups. In particular, we hope that our discussion better emphasizes that this work merely takes a first step, and that we do not claim to make a "large" breakthrough at this point.

---

## Round 2 · Referee Report · Anonymous · 2022-4-20

Report
I thank the authors for thoroughly addressing the points made in my previous report. I think the newly-modified introduction is much improved and does a very good job of discussing many of the open issues in the recent developments of black hole evaporation, including with 2D models, nongravitating baths, and doubly-holographic/braneworld constructions. Unfortunately, I must admit that I don't think that these changes substantially address one of my most pressing concerns: namely, of all the ways one could modify the bath, what makes turning on a relevant deformation special? Currently the paper does a very good job of motivating the //general// need to understand properties of the bath and how it affects the physics of the evaporation process, but I don't think it substantially addresses how the //specific// modification being considered here makes headway in addressing this goal. What lesson has been learned? Or if this paper is a first step, what lesson(s) will hopefully be learned in subsequent work as a consequence of studying these sorts of deformations? (To be frank, my current reading of the paper is that a relevant deformation is considered for no other reason than that it's something one can do, which I assume means I'm still missing the authors' physical motivation.)
The authors do mention a potential answer to some of these questions in their response to my first report: that "In the end, our results show that the bath is not a consistent computational tool." What does it mean to be an inconsistent computational tool? Is one of the points of the paper is that models involving baths do not accurately capture the relevant physics of the evaporation process? If so, this would be an very interesting and impactful claim to make. I did not get the sense that this was one of the lessons from reading the paper, but if it is, I think the paper would benefit substantially from making this point much more explicit.
With its current modifications, I think the paper might be appropriate to publish in certain journals as an exploration into possible modifications of the bath using a doubly-holographic setup. However, SciPost's stated acceptance criteria are quite high, and unfortunately without addressing the questions I posed above I don't think the new version of the paper meets them.

---

## Round 2 · Author Response

List of changes
- More discussion added to Introduction
- Introduction reformatted into subsections
- Additional future directions based on referee remarks added

You are currently on this page

---

## Round 2 · List of Changes

- More discussion added to Introduction
- Introduction reformatted into subsections
- Additional future directions based on referee remarks added

You are currently on this page

---

## Editorial Decision

published